# Towards Understanding Massive Activations in Attention Sink Mechanism

**Haiyu Wang** [1]   **Yuanyuan Lin** [1]

## Abstract

Recent studies have revealed two intriguing phenomena in large language models: attention sinks and massive activations. However, the co-emergence and co-existence of these two phenomena remain poorly understood. In this work, we revisit the prevailing view that massive activations are the primary mechanism responsible for concentrating attention on sink tokens, and provide a more nuanced interpretation of their relationship. Through both theoretical analysis and empirical evidence, we demonstrate that massive activations and attention sinks jointly act to prevent excessive token mixing in self-attention. Specifically, attention sinks suppress mixing among non-sink tokens, whereas massive activations suppress mixing between sink tokens and non-sink tokens. Furthermore, our theory provides a principled explanation of how KV-biases, gating mechanisms and normalization layers can remove massive activations while largely preserving attention sinks. We further conduct intervention analyses and find that removing the value vector of the sink token can recover attention sinks even when massive activations are entirely suppressed. Overall, this work provides a mechanistic perspective on how massive activations and attention sinks interact under normalization and self-attention layers, offering new insights into their functional roles in Transformer models. Code is available at Attn-MA.

## 1. Introduction

Large language models have sparked transformative advances across a wide range of fields; however, our understanding of their internal mechanisms remains limited. Among the many unintuitive yet prevalent phenomena observed in LLMs, two have attracted particularly significant attention from researchers. In particular, Xiao et al. (2023) showed that LLMs consistently assign disproportionately large attention weights to the initial tokens, regardless of their semantic relevance. This phenomenon, known as the *attention sinks*, has enabled a broad range of applications across different domains (Xiao et al., 2023; Han et al., 2024; Yang et al., 2025; Ge et al., 2023; Wan et al., 2024; Wu & Tu, 2024; Zhang et al., 2024; Chen et al., 2024; Liu et al., 2024; Huang et al., 2024). Another puzzling phenomenon is the emergence of massive activations. By examining the hidden states of various LLMs, Sun et al. (2024) found that certain activations exhibit huge magnitudes in the hidden states of sink tokens (the first token, delimiter tokens and certain tokens with weak semantics), making the norm of their hidden states larger than non-sink tokens.

Interestingly, a number of prior works have observed that attention sinks tend to co-occur with massive activations. Specifically, Gu et al. (2024); Sun et al. (2024) reported that once massive activations emerge at a certain layer, attention in subsequent layers becomes highly concentrated on tokens associated with these activations. To explain this phenomenon, Sun et al. (2024) suggested that large language models employ massive activations to concentrate a substantial amount of attention on sink tokens. Moreover, Gu et al. (2021) hypothesized that after RMSNorm, the magnitudes of massive activations are largely preserved, while the remaining activations are suppressed. As a result, the query, key, and value vectors are effectively distributed on different manifolds, which causes the key of the sink tokens to be closer to the queries of non-sink tokens. More recently, Queipo-de Llano et al. (2025) used massive activations as a unifying factor linking compression valleys (Skean et al., 2025) and attention sink. They show that when the beginning-of-sequence token develops extremely large activation norms in intermediate layers, compression valleys and attention sinks emerge simultaneously.

Despite these observations, existing works do not fully explain why massive activations systematically co-occur with attention sinks, why they predominantly emerge on sink tokens, or why they disappear in certain architectures

[1]Department of Statistics and Data Science, The Chinese University of Hong Kong, Hong Kong. Correspondence to: Haiyu Wang <HaiYuWang@link.cuhk.edu.hk>, Yuanyuan Lin <ylin@sta.cuhk.edu.hk>.

*Proceedings of the 43rd International Conference on Machine Learning*, Seoul, South Korea. PMLR 306, 2026. Copyright 2026 by the author(s).

while attention sink persists. The existing explanation can not explain even if RMSNorm is scaling-invariant, the norm of the hidden states of sink tokens are still much larger than non-sink tokens. Given the consistent co-occurrence of massive activations and attention sinks, a deeper understanding of their relationship is therefore essential.

In this work, we aim to clarify the relationship between massive activations and attention sinks. From an empirical perspective, we investigate the underlying mechanism of attention sinks and find that massive activations are not necessary for their formation. Instead, they play a crucial role in preserving the attention sink structure across layers, and improper interventions on massive activations can disrupt this structure and degrade model performance. From a theoretical perspective, we show that massive activations naturally emerge alongside attention sinks when attention heads implicitly act to prevent over-mixing (Barbero et al., 2025). In particular, the hidden states of sink tokens tend to develop large norms to mitigate mixing between sink and non-sink tokens, which arises as a consequence of the Softmax attention and the pre-LN architecture adopted in most large language models.

Our contributions are summarized as follows:

- Building on the notion of over-mixing introduced in (Barbero et al., 2025), we provide a theoretical analysis showing that when attention is highly concentrated on sink tokens, massive activations at the sink tokens are necessary to limit mixing between sink and non-sink tokens.

- Leveraging our theoretical framework, we explain the placement of massive activations in LLMs with post-LN architectures (Devlin et al., 2019), and why KV biases (Sun et al., 2024; Gu et al., 2024), gating mechanisms (Bondarenko et al., 2023; An et al., 2025; Qiu et al., 2025), activation functions without normalization, DyT (Zhu et al., 2025; Owen et al., 2025) can effectively mitigate massive activations.

- We conduct extensive experiments demonstrating that massive activations are not necessary for the formation of attention sinks in pre-trained large language models. We further analyze the mechanisms by which eliminating massive activations weakens attention sinks and degrades model performance.

## 2. Related Works

**Attention sinks.** The attention sink phenomenon first termed by (Xiao et al., 2023) has been widely observed in LLMs and vision Transformers (Han et al., 2024; Darcet et al., 2023; Kang et al., 2025) with its importance being confirmed by many works in different fields (Hooper et al., 2024; Son et al., 2024; Su & Yuan, 2025). Cancedda (2024) showed, from a spectral perspective, that specific subspaces are responsible for the creation of attention sink. Guo et al. (2024) provided empirical and theoretical analyses of attention sinks in small Transformer LMs on the Bigram-Backcopy task. Gu et al. (2024) conducted extensive experiments to investigate how optimization, data distribution, loss function, and model architecture will influence the emergence of attention sinks. Barbero et al. (2025) argued that attention sink mechanism provides a method for LLMs to avoid over-mixing, connecting this to existing lines of work that study mathematically how information propagates in Transformers. Sun et al. (2024) observed that sink tokens are not limited to the starting token but also some delimiter tokens and word tokens with weak semantics. Yu et al. (2024) found that not all attention sinks have a positive impact on the achievable accuracy of LLMs. Specifically, for the majority of attention sinks occurring in the middle or later parts of inputs, reducing their attention scores can result in improved accuracy. Wong et al. (2025) found the existence of secondary sinks that arise primarily in middle layers and can persist for a variable number of layers, and draw a smaller, but still significant, amount of attention mass.

**Massive activations.** Sun et al. (2024) found that LLMs tend to allocate substantial attention to tokens that are assigned with massive activations and setting merely four massive activations (out of millions of activations) play a critical role in LLM's capabilities. Queipo-de Llano et al. (2025) connected compression valley and attention sinks by massive activations, showing that when the beginning-of-sequence token develops extreme activation norms in the middle layers, both compression valleys and attention sink emerge simultaneously. Oh et al. (2024) showed that the massive activations come from several massive weights in models and they explored the relationship between massive activations and the position of the bos token. Yona et al. (2025) showed that long sequences of repeated token may also exhibit massive activations. Gallego-Feliciano et al. (2025) presented a comprehensive analysis of massive activations development throughout training process, demonstrating that their emergence follows predictable mathematical patterns.

## 3. Preliminaries

Let $f_\theta$ be an auto-regressive LM with $L$ transformer decoder blocks and $X \in \mathbb{R}^{|\mathbb{V}| \times T} := [x_1, \cdots, x_T]$ are the input tokens, where each $x_i$ represents a one-hot encoding and $|\mathbb{V}|$ is the vocabulary size of tokenizer $\mathbb{V}$. The LM output is also a sequence $Y \in \mathbb{R}^{|\mathbb{V}| \times T} := [y_1, \cdots, y_T] = f_\theta(X)$, where each $y_i$ represents the predicted logits of $p(x_{t+1}|x_{\leq t})$.

In what follows, we present the basic concepts of the Transformer architecture, largely following the notation in (Gu et al., 2024).

**Transformer blocks.** For the forward pass, $\boldsymbol{X}$ is first embedded as $\boldsymbol{H}^0 \in \mathbb{R}^{d \times T} := \boldsymbol{W}_E \boldsymbol{X} + \boldsymbol{P}$, where $d$ is the embedding size, $\boldsymbol{W}_E \in \mathbb{R}^{d \times |\mathbb{V}|}$ is the learnable word embedding matrix, $\boldsymbol{P} \in \mathbb{R}^{d \times T}$ is the position embedding. We denote $\boldsymbol{H}^l \in \mathbb{R}^{d \times T} := [\boldsymbol{h}_1^l, \boldsymbol{h}_2^l, \cdots, \boldsymbol{h}_T^l], 1 \leq l \leq L$ to be the output of the $l$-th block. Each block comprises a multi head self-attention self-attention (MHSA) operation and a feed-forward network (FFN). The block has either a pre-norm or a post-norm structure according to the location of layer normalization (LN) (Ba et al., 2016; Zhang & Sennrich, 2019). Most of LLMs adopt a pre-norm structure and we mainly focus on RMSNorm operation in this work

$$\boldsymbol{H}^\ell = \text{FFN}(\text{LN}(\boldsymbol{H}^{\ell-1/2})) + \boldsymbol{H}^{\ell-1/2},$$
$$\boldsymbol{H}^{\ell-1/2} = \boldsymbol{O}^{\ell-1} + \boldsymbol{H}^{\ell-1},$$
$$\boldsymbol{O}^{\ell-1} = \text{MHSA}(\text{LN}(\boldsymbol{H}^{\ell-1})).$$

**MHSA layers.** In the MHSA layer, the input $\boldsymbol{H}^{l-1}$ are first transformed into keys, queries, and values: $\boldsymbol{K}^{l,h} = \boldsymbol{W}_K^{l,h} \text{LN}(\boldsymbol{H}^{l-1})$, $\boldsymbol{Q}^{l,h} = \boldsymbol{W}_Q^{l,h} \text{LN}(\boldsymbol{H}^{l-1})$, $\boldsymbol{V}^{l,h} = \boldsymbol{W}_V^{l,h} \text{LN}(\boldsymbol{H}^{l-1})$ for each head $1 \leq h \leq H$, where $\boldsymbol{W}_K^{l,h}, \boldsymbol{W}_Q^{l,h}, \boldsymbol{W}_V^{l,h} \in \mathbb{R}^{d_h \times d}$ with $d_h = d/H$. Then the attention output is computed as

$$\boldsymbol{A}^{l,h} = \text{Softmax}\left((\boldsymbol{K}^{l,h})^\top \boldsymbol{Q}^{l,h}/\sqrt{d_h} + \boldsymbol{M}\right), \quad (1)$$
$$\boldsymbol{O}^l = \boldsymbol{W}_O^l \text{Concat}_{h=1}^H(\boldsymbol{V}^{l,h} \boldsymbol{A}^{l,h}), \quad (2)$$

where $\boldsymbol{W}_O^l \in \mathbb{R}^{d \times d}$ $\boldsymbol{M} \in \mathbb{R}^{T \times T}$ is an attention mask. For vanilla causal attention, $\boldsymbol{M}_{ij} = -\infty$ for $i > j$ and $\boldsymbol{M}_{ij} = 0$ for $i \leq j$. In our setting, $\boldsymbol{A}^{\ell,h}$ is an upper triangular matrix and Softmax function is applied to each column. Finally, the output of final Transformer block $\boldsymbol{H}^L$ is fed into an unembedding layer for prediction: $\boldsymbol{Y} = \boldsymbol{W}_{\text{cls}} \text{LN}(\boldsymbol{H}^L)$, and $\boldsymbol{W}_{\text{cls}} \in \mathbb{R}^{|\mathbb{V}| \times d}$.

**Attention sinks.** Attention sinks represent the phenomenon where certain sink tokens are assigned with significant attention scores. In our theoretical analysis, we focus primarily on the case where the sink token is the first token, which is the most common setting in practice. To measure the presence of the sink, we follow the metric proposed by (Gu et al., 2024): $\text{Sink}_1^\varepsilon = \frac{1}{LH} \sum_{\ell,h} \mathbf{1}(\frac{1}{T} \sum_{i=1}^T A_{1,i}^{\ell,h} > \varepsilon)$. For $T = 64$, we set $\varepsilon = 0.3$.

## 4. Theoretical Analysis

In the following, we present a theoretical result that explains why the sink tokens tend to acquire a large norm as

the attention sinks emerge. Furthermore, we leverage our theoretical framework to account for several existing phenomena. Our analysis builds on the notion proposed in (Barbero et al., 2025) that attention sinks serve as a mechanism for preventing over-mixing in large language models.

### 4.1. Main Results

Following the basic idea in (Barbero et al., 2025), we consider using $\left\|\frac{\partial \boldsymbol{H}^L}{\partial \boldsymbol{H}^0}\right\|$ to quantify the degree of mixing by the spectral norm, which measures the sensitivity of the representation at layer $L$ is to perturbations in the input $\boldsymbol{H}^0$. By the chain rule, this quantity can be expressed as

$$\frac{\partial \boldsymbol{H}^L}{\partial \boldsymbol{H}^0} = \frac{\partial \boldsymbol{H}^L}{\partial \boldsymbol{H}^{L-1}} \cdots \frac{\partial \boldsymbol{H}^1}{\partial \boldsymbol{H}^0}.$$

Recall the definition of a Transformer block:

$$\boldsymbol{H}^{\ell+1} = \text{FFN}\left(\text{LN}\left(\boldsymbol{H}^{\ell+1/2}\right)\right) + \boldsymbol{H}^{\ell+1/2},$$
$$\boldsymbol{H}^{\ell+1/2} = \boldsymbol{O}^{\ell+1} + \boldsymbol{H}^\ell,$$
$$\boldsymbol{O}^{\ell+1} = \text{MHSA}(\text{LN}(\boldsymbol{H}^\ell)).$$

The corresponding Jacobian can therefore be written as

$$\frac{\partial \boldsymbol{H}^{\ell+1}}{\partial \boldsymbol{H}^\ell} = \frac{\partial \boldsymbol{H}^{\ell+1}}{\partial \boldsymbol{H}^{\ell+1/2}} \frac{\partial \boldsymbol{H}^{\ell+1/2}}{\partial \boldsymbol{H}^\ell},$$

and the exact form of $\frac{\partial \boldsymbol{H}^{\ell+1}}{\partial \boldsymbol{H}^{\ell+1/2}}$ and $\frac{\partial \boldsymbol{H}^{\ell+1/2}}{\partial \boldsymbol{H}^\ell}$ are

$$\frac{\partial \boldsymbol{H}^{\ell+1}}{\partial \boldsymbol{H}^{\ell+1/2}} = \frac{\partial \text{FFN}(\text{LN}(\boldsymbol{H}^{\ell+1/2}))}{\partial \text{LN}(\boldsymbol{H}^{\ell+1/2})} \frac{\partial \text{LN}(\boldsymbol{H}^{\ell+1/2})}{\partial \boldsymbol{H}^{\ell+1/2}} + \boldsymbol{I},$$
$$\frac{\partial \boldsymbol{H}^{\ell+1/2}}{\partial \boldsymbol{H}^\ell} = \frac{\partial \text{MHSA}\left(\text{LN}(\boldsymbol{H}^\ell)\right)}{\partial \text{LN}\left(\boldsymbol{H}^\ell\right)} \frac{\partial \text{LN}(\boldsymbol{H}^\ell)}{\partial \boldsymbol{H}^\ell} + \boldsymbol{I},$$

where we use $\boldsymbol{I}$ to denote $\frac{\partial \boldsymbol{H}^{\ell+1/2}}{\partial \boldsymbol{H}^{\ell+1/2}}$ and $\frac{\partial \boldsymbol{H}^\ell}{\partial \boldsymbol{H}^\ell}$. Using the sub-multiplicativity of the spectral norm, we have

$$\left\|\frac{\partial \boldsymbol{H}^L}{\partial \boldsymbol{H}^0}\right\| \leq \left\|\frac{\partial \boldsymbol{H}^L}{\partial \boldsymbol{H}^{L-1}}\right\| \cdots \left\|\frac{\partial \boldsymbol{H}^1}{\partial \boldsymbol{H}^0}\right\|.$$

Consequently, bounding $\left\|\frac{\partial \boldsymbol{H}^L}{\partial \boldsymbol{H}^0}\right\|$ reduces to bounding each layer-wise term $\left\|\frac{\partial \boldsymbol{H}^{\ell+1}}{\partial \boldsymbol{H}^\ell}\right\|$. Since the FFN operates independently on each token, we focus on deriving an upper bound on $\left\|\frac{\partial \boldsymbol{H}^{\ell+1/2}}{\partial \boldsymbol{H}^\ell}\right\|$ where attention sinks occur. Using the inequality $\left\|\frac{\partial \boldsymbol{H}^{\ell+1/2}}{\partial \boldsymbol{H}^\ell}\right\| \leq \sum_{i=1}^T \left\|\frac{\partial \boldsymbol{H}^{\ell+1/2}}{\partial \boldsymbol{h}_i^\ell}\right\|$ and to distinguish the different roles played by the sink token and non-sink tokens, we separately bound $\left\|\frac{\partial \boldsymbol{H}^{\ell+1/2}}{\partial \boldsymbol{h}_1^\ell}\right\|$ and $\left\|\frac{\partial \boldsymbol{H}^{\ell+1/2}}{\partial \boldsymbol{h}_i^\ell}\right\|$ for $i \neq 1$.

In the following theorem, we show that the presence of attention sinks is potential to enlarge $\left\|\frac{\partial \boldsymbol{H}^{\ell+1/2}}{\partial \boldsymbol{h}_1^\ell}\right\|$, whereas

a large norm of the first (sink) token reduces this quantity, thereby helping mitigate over-mixing. For generality, we do not consider causal mask in the following theorem.

**Theorem 4.1.** *Let the embedding dimension $d \geq 1$ and the sequence length $T \geq 2$ be fixed. Suppose there exist constants $\{\varepsilon_i\}_{i=1}^T \subset \mathbb{R}$ and $\epsilon, \epsilon' \in (0,1)$ such that*

$$0 < \epsilon \leq \varepsilon_i \leq \varepsilon' < 1,$$

*and in the $(\ell+1)$-th self-attention layer, the attention probability matrix satisfies*

$$\boldsymbol{A}_{1,i}^{\ell+1,h} = 1 - \varepsilon_i, \quad \forall i \in [T], \forall h \in [H].$$

*Then the following bound holds:*

$$\left\| \frac{\partial \boldsymbol{H}^{\ell+1/2}}{\partial \boldsymbol{h}_i^\ell} \right\| \leq \begin{cases} K_1 \frac{\sqrt{T}(1-\varepsilon)}{\|\boldsymbol{h}_1^\ell\|} + K_2 \frac{\varepsilon'}{\|\boldsymbol{h}_1^\ell\|} + 1, & i = 1, \\ \\ K_1 \frac{(\sqrt{T-1})^{-1}\varepsilon'}{\|\boldsymbol{h}_i^\ell\|} + K_2 \frac{\varepsilon'}{\|\boldsymbol{h}_i^\ell\|} + 1, & i \neq 1, \end{cases}$$

*where $K_1, K_2 \in \mathbb{R}$ depend only on $d, H$, and model parameters.*

**Meaning of the theorem.** A detailed proof of Theorem 4.1, along with a discussion of the assumptions, is provided in Appendix B.

- The constant 1 originates from the skip connection and is independent of $\|\boldsymbol{h}_i^\ell\|$. This component provides a non-vanishing baseline for gradient propagation.

- $K_1 \frac{\sqrt{T}(1-\varepsilon)}{\|\boldsymbol{h}_1^\ell\|}$ and $K_1 \frac{\varepsilon'}{\|\boldsymbol{h}_i^\ell\|}$ quantify the sensitivity of the output with respect to $\boldsymbol{h}_1^\ell$ and $\boldsymbol{h}_{i\neq1}^\ell$ through the value path. Compared to non-sink tokens, the interaction between the first token and the remaining tokens dominates the overall token mixing, since most tokens attend predominantly to the first one. This behavior highlights the necessity of massive activations in regulating such interactions. This effect is realized by the RMSNorm applied before each self-attention module, which introduces a factor of order $\|\boldsymbol{h}_i^\ell\|^{-1}$ to the Jacobian norm bound. In particular, when $\|\boldsymbol{h}_1^\ell\|$ is sufficiently large and the perturbation $\|\Delta\boldsymbol{h}_1^\ell\|$ is small, RMSNorm significantly attenuates the effect of perturbations:

$$\left\| \text{LN}(\boldsymbol{h}_1^\ell + \Delta\boldsymbol{h}_1^\ell) - \text{LN}(\boldsymbol{h}_1^\ell) \right\| \approx 0.$$

- The term $K_2 \frac{\epsilon'}{\|\boldsymbol{h}_i^\ell\|}$ captures the sensitivity respect to $\boldsymbol{h}_i^\ell$ through the path of attention weights, which remains well controlled for both $\boldsymbol{h}_1^\ell$ and $\boldsymbol{h}_{i\neq1}^\ell$, even without massive activations.

Taken together, these results indicate that the coexistence of small $\frac{\varepsilon'}{\sqrt{T-1}}$ (attention sink) and large $\|\boldsymbol{h}_1^\ell\|$ (massive activations) implicitly regularizes self-attention layers by preventing excessive mixing across tokens. Intuitively, attention sink halts the mixing among $\boldsymbol{h}_2^\ell, \cdots, \boldsymbol{h}_T^\ell$, while the massive activations further inhibit mixing between the sink token $\boldsymbol{h}_1^\ell$ and the remaining tokens through RMSNorm-induced insensitivity.

**Effects of FFNs.** Theorem 4.1 intentionally isolates the self-attention mechanism, as this is where the attention sink phenomenon fundamentally operates. FFNs require distinct theoretical treatment; as detailed in Appendix D, FFNs amplify massive activations by potentially perturbing the residual stream, thereby intensifying token mixing independently of the attention layers.

**Massive activations induce large norm.** As shown in Theorem 4.1, the role of massive activations is to induce a large $\|\boldsymbol{h}_1^\ell\|$, which in turn leads to a small $\left\| \frac{\partial \boldsymbol{H}^{\ell+1/2}}{\partial \boldsymbol{h}_1^\ell} \right\|$. However, the observations in Sun et al. (2024) indicate that massive activations are highly sparse within the hidden states of sink tokens. Notably, such activations do not need to be sparse in order to produce a large hidden-state norm. Gu et al. (2024) hypothesized that, after RMSNorm, the coordinates corresponding to massive activations are preserved while the remaining coordinates are effectively suppressed, thereby playing a key role in the formation of attention sinks. Nevertheless, as we show in Section 6, these massive-activation coordinates may not be as crucial as previously believed. Moreover, Park et al. (2025) trained a family of LLMs in which massive activations are distributed uniformly across the hidden states of sink tokens. This observation suggests that the sparsity of massive activations may arise from optimization methods or other architectural choices, which are beyond the scope of this work. Theorem 4.1 primarily establishes that the norm of sink-token hidden states must be sufficiently large, despite the approximate scaling invariance of RMSNorm. A more detailed discussion is provided in Appendix C.

### 4.2. Empirical Validation

This section presents an empirical validation of the theoretical insights established in Theorem 4.1. Our primary focus is twofold: assessing the capacity of massive activations to mitigate token mixing, and evaluating the scaling behavior of the Jacobian norm with respect to $\|\boldsymbol{h}_i^\ell\|$ and $\varepsilon'$.

**Token mixing.** To quantify the degree of token mixing in real-world models, we first track how perturbations introduced at the sink and non-sink tokens affect the hidden representations of the final token, observing how this

influence evolves across layers. To isolate the stabilizing role of massive activations, we zero them out while roughly preserving the structure of the attention sinks. As illustrated in Figure 1, the propagation of these perturbations becomes significantly amplified in the absence of massive activations.

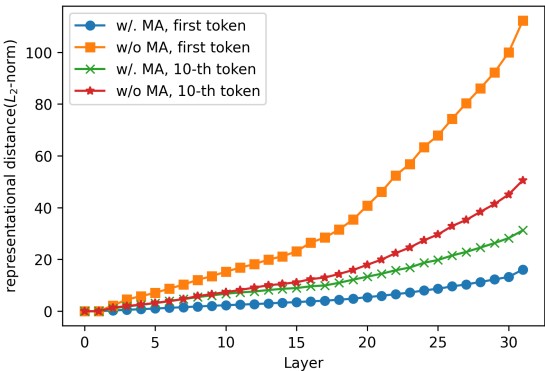

*Figure 1.* **Effect of token perturbations on Llama2-7B.** We report the representational distances induced by token perturbations, comparing settings with or without massive activations.

To further investigate how RMSNorm, in conjunction with massive activations, suppresses mixing from sink tokens, we perturb the hidden states both before and after the application of RMSNorm and track the resulting changes in the output. As illustrated in Figure 2, when perturbations are injected prior to RMSNorm, the output is least sensitive to the first token; conversely, when perturbations are introduced after RMSNorm, the first token becomes the most sensitive.

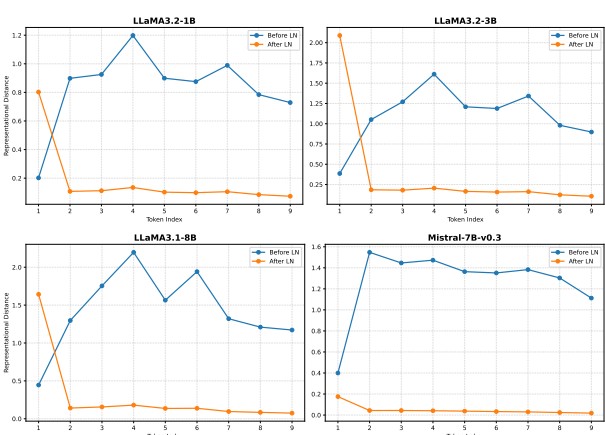

*Figure 2.* Effect of token perturbations before and after RMSNorm.

**Estimating** $\left\|\frac{\partial \boldsymbol{H}^{\ell+1/2}}{\partial \boldsymbol{h}_1^\ell}\right\|$**.** As established in Theorem 4.1, the upper bound of $\left\|\frac{\partial \boldsymbol{H}^{\ell+1/2}}{\partial \boldsymbol{h}_1^\ell}\right\|$ scales inversely with $\|\boldsymbol{h}_1^\ell\|$. To empirically verify this, we enforce $\epsilon = \epsilon'$ by rearranging the attention weights in a pre-trained self-attention layer. We then rescale $\|\boldsymbol{h}_1^\ell\|$ to $\alpha \frac{\boldsymbol{h}_1^\ell}{\|\boldsymbol{h}_1^\ell\|}$ for $\alpha \in \{0.1, 1, 10, 100, 1000\}$.

Notably, due to the scale-invariance of RMSNorm, this operation does not alter the final outputs. As illustrated in Figure 3, $\left\|\frac{\partial \boldsymbol{H}^{\ell+1/2}}{\partial \boldsymbol{h}_1^\ell}\right\|$ decreases monotonically as $\|\boldsymbol{h}_1^\ell\|$ increases, corroborating our theoretical bound.

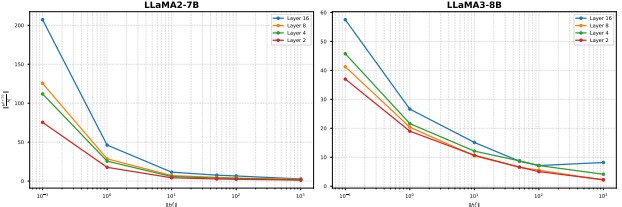

*Figure 3.* Estimation of $\|\partial \boldsymbol{H}^{\ell+1/2}/\partial \boldsymbol{h}_1^\ell\|$.

**Estimating** $\left\|\frac{\partial \boldsymbol{H}^{\ell+1/2}}{\partial \boldsymbol{h}_i^\ell}\right\|$**.** To estimate $\left\|\frac{\partial \boldsymbol{H}^{\ell+1/2}}{\partial \boldsymbol{h}_i^\ell}\right\|$ for non-sink tokens $(i \neq 1)$, it is necessary to isolate the effect of attention distribution from the norm of token embeddings. To achieve this, we normalize the hidden states such that $\|\boldsymbol{h}_i^\ell\| = 1$. According to the theoretical formulation in Theorem 4.1, the upper bound of this Jacobian norm scales monotonically with $\varepsilon'$, which represents the attention weights allocated to the non-sink tokens. Figure 4 empirically corroborates this bound across different layers. As $\varepsilon'$ decreases (a condition indicating that the attention sink is highly active), the layer's sensitivity to the $i$-th token drops correspondingly. Mechanically, this confirms that attention sink suppresses the token mixing among non-sink tokens.

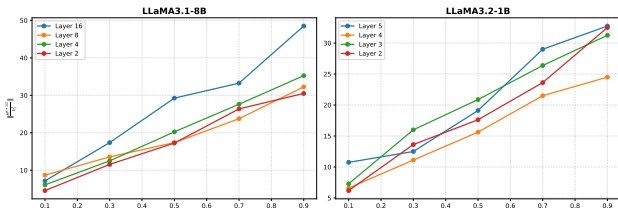

*Figure 4.* Estimation of $\|\partial \boldsymbol{H}^{\ell+1/2}/\partial \boldsymbol{h}_i^\ell\|$.

### 4.3. Explanation of Existing Phenomena

In this section, we try to answer (i) why KV biases, gating mechanisms, self-attention without Softmax and specialized layer normalization can alleviate massive activations, (ii) how the placement of layer normalization dictates the position of massive activations, (iii) why massive activations emerge in secondary sink tokens, (iv) why can the keys and values of the sink tokens consistently maintain exceptionally low norms.

Theorem 4.1 demonstrates that the necessity for massive activations stems from an amplified sensitivity to the sink token, bounded by $K_1 \frac{\sqrt{T}(1-\varepsilon)}{\|\boldsymbol{h}_1^\ell\|}$, where $\sqrt{T}(1-\varepsilon)$ comes

from Softmax which forces the sum of all attention scores to be 1, $K_1$ largely depends on $\|\boldsymbol{W}_O\|\|\boldsymbol{W}_V\|$ which is shared across tokens then is hard to be trained to be small, and $\frac{1}{\|\boldsymbol{h}_1^\ell\|}$ originates from RMSNorm. Naturally, if the coefficient $K_1\sqrt{T}(1-\varepsilon)$ can be largely reduced or this term decays at a much higher order with respect to $\|\boldsymbol{h}_1^\ell\|$, then massive activations may not be required. Based on this analysis, we arrive at the following conclusions. See a detailed discussion in Appendix F.

**KV biases eliminating $\frac{K_1\sqrt{T}(1-\varepsilon)}{\|\boldsymbol{h}_1^\ell\|}$.** KV biases $\boldsymbol{k}^{*\ell,h}$, $\boldsymbol{v}^{*\ell,h}$ proposed in (Sun et al., 2024; Gu et al., 2024) are directly learned KV vectors as model parameters rather than being computed from hidden states (i.e., $\boldsymbol{k}^{*\ell,h} \neq \boldsymbol{W}_K^{\ell,h} \mathrm{LN}(\boldsymbol{h}_1^\ell)$ and $\boldsymbol{v}^{*\ell,h} \neq \boldsymbol{W}_V^{\ell,h} \mathrm{LN}(\boldsymbol{h}_1^\ell)$). As a result, gradients that would carry signals inducing massive activations are no longer propagated to $\boldsymbol{h}_1^\ell$. Therefore, when attention scores are largely dominated by $\boldsymbol{k}^{*\ell,h}$, massive activations are not expected to emerge. However, if $\boldsymbol{k}^{*\ell,h}$ fails to attract most attention scores or if multiple sink tokens are present, massive activations may still occur. Intuitively, the success of KV biases to eliminate massive activations relies on where they can attract most attention weights and no other sink tokens will appear.

**Value projection gating mechanism reduces $K_1$.** Qiu et al. (2025) showed that introducing a gating mechanism solely on value projection in self-attention can alleviate massive activations while preserving attention sinks. $K_1$ mostly depends on $\boldsymbol{W}_V$ and $\boldsymbol{W}_O$. A small $K_1$ rarely happens since parameters are shared with all tokens. After applying value gating mechanism, $K_1 = O(\|\boldsymbol{W}_O\|\|\boldsymbol{W}_V\| \cdot s)$, where $s$ is the input-dependent gating score corresponding to the sink tokens, then $K_1$ can be greatly reduced by a small $s$. This is partly verified by the observations (Qiu et al., 2025) that these gating scores are highly sparse and close to zero.

**Attention output gating mechanism reduces $\sqrt{T}(1-\varepsilon)$.** Bondarenko et al. (2023); An et al. (2025); Qiu et al. (2025) demonstrated that applying a gating mechanism to attention output can simultaneously alleviate attention sinks and massive activations. This mechanism allows individual attention output to be nearly zero without forming attention sinks, thereby reducing the factor $\sqrt{T}(1-\varepsilon)$ caused by most attentions weights are assigned with the sink tokens, which alleviates massive activations.

**Activation functions without normalization reduces $\sqrt{T}(1-\varepsilon)$.** Gu et al. (2024) showed replacing the Softmax function in self-attention with unnormalizaed alternatives (i.e., sigmoid, elu, mlp, inner product) leads to the disappearance of both attention sink and massive activations. Similar to attention output gating, this modification allows for more flexible attention score distributions. Since sink tokens no longer receive disproportionally large attention

scores, massive activations are unlikely to emerge.

**Special layer normalization changes $\|\boldsymbol{h}_1^\ell\|^{-1}$.** Zhu et al. (2025) proposed a Transformer architecture without normalization, in which each layer normalization is replaced by a DyT module. In Appendix F, we will show that massive activations are not required in this architecture, as DyT induces an exponential decay with respect to each coordinate of $\boldsymbol{h}_1^\ell$. This behavior is consistent with the empirical observations reported in (Owen et al., 2025).

**Massive activations in post-LN structure.** By replacing the pre-LN to post-LN, Gu et al. (2024) observed that massive activations emerge in the hidden states before LN, rather than in $\boldsymbol{h}_1^\ell$. Although architectures equipped with post-LN are not widely adopted in contemporary large language models, they also exhibit the attention sink phenomenon (Clark et al., 2019; Kobayashi et al., 2020). This setting provides an additional opportunity to verify our theoretical analysis. Recall that the post-LN structure can be written as follows:

$$\boldsymbol{H}^{\ell+1} = \mathrm{LN}\left(\mathrm{FFN}(\mathrm{LN}(\boldsymbol{H}^{\ell+1/2})) + \mathrm{LN}(\boldsymbol{H}^{\ell+1/2})\right),$$

$$\boldsymbol{H}^{\ell+1/2} = \mathrm{MHSA}(\boldsymbol{H}^\ell) + \boldsymbol{H}^\ell.$$

Let $\boldsymbol{H}^* = \mathrm{FFN}(\mathrm{LN}(\boldsymbol{H}^{\ell-1/2})) + \mathrm{LN}(\boldsymbol{H}^{\ell-1/2})$, then we have $\boldsymbol{H}^\ell = \mathrm{LN}(\boldsymbol{H}^*)$ and

$$\frac{\partial \boldsymbol{H}^{\ell+1/2}}{\partial \boldsymbol{H}^*} = \frac{\partial \mathrm{MHSA}(\boldsymbol{H}^\ell)}{\partial \boldsymbol{H}^\ell}\frac{\partial \boldsymbol{H}^\ell}{\partial \boldsymbol{H}^*} + \frac{\partial \boldsymbol{H}^\ell}{\partial \boldsymbol{H}^*}$$
$$= \frac{\partial \mathrm{MHSA}(\mathrm{LN}(\boldsymbol{H}^*))}{\partial \mathrm{LN}(\boldsymbol{H}^*)}\frac{\partial \mathrm{LN}(\boldsymbol{H}^*)}{\partial \boldsymbol{H}^*} + \frac{\partial \mathrm{LN}(\boldsymbol{H}^*)}{\partial \boldsymbol{H}^*},$$

which implies that $\boldsymbol{H}^{\ell+1/2}$ is connected to $\boldsymbol{H}^*$ through a pre-LN structure rather than to $\boldsymbol{H}^\ell$. Consequently, massive activations are expected to arise in $\boldsymbol{H}^*$, i.e., the input to the second LN module. As shown in Figure 5, we plot the last column of $\boldsymbol{H}^*$ corresponding to the hidden state of [SEP] token in BERT (Devlin et al., 2019) and observe that the massive activations persist across layers.

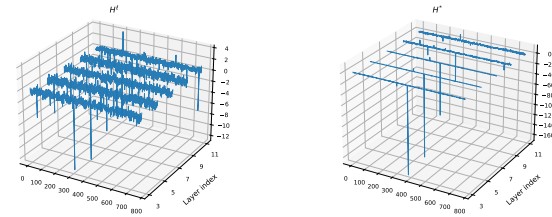

*Figure 5.* Massive activations are in $\boldsymbol{H}^*$ instead of $\boldsymbol{H}^\ell$ in BERT.

**More than one token with massive activations.** Prior empirical studies indicate that sink tokens are not limited to [BOS]; other tokens may also function as sinks. In particular, Sun et al. (2024) showed that starting tokens

besides `[BOS]`, as well as semantically uninformative delimiter tokens, can attract a disproportionate amount of attention and thus behave as sink tokens. These tokens share a common characteristic in that they provide little semantic content while occupying structurally salient positions in the sequence.

Theorem 4.1 naturally extends to settings with multiple sink tokens. In this case, the Jacobian norm is jointly influenced by all sink tokens, and achieving a relatively small upper bound on the Jacobian norm requires the hidden states of all sink tokens to have sufficiently large norms. Intuitively, when multiple tokens attract dominant attention mass, the sensitivity of the self-attention output becomes distributed across these sinks. To counteract the resulting amplification through the normalization in Softmax, each sink token must exhibit massive activations so that subsequent normalization layers can effectively dampen perturbations.

**Massive activations make keys and values stable.** Let $\ell_0$ denote the layer at which massive activations first appear in the input, and $\ell_1$ the layer at which they disappear at the output. By Theorem 4.1, the outputs of the FFN and self-attention corresponding to the first token can be viewed as small perturbations to its hidden state based on the fact that the first token is not influenced by other tokens due to the causal mask, so it can be analyzed independently. Prior work has shown that the norm $\|\boldsymbol{v}_1^\ell\|$ is small compared to those of non-sink tokens (Gu et al., 2024; Guo et al., 2024), and that the FFN output norm is also small after massive activations emerge (Cancedda, 2024). Specifically, when $\|\operatorname{Attn}(\boldsymbol{h}_1^{\ell_0})\| = \|\boldsymbol{v}_1^{\ell_0}\|$ is small, then $\operatorname{LN}(\boldsymbol{h}_1^{\ell_0+1/2}) \approx \operatorname{LN}(\boldsymbol{h}_1^{\ell_0})$. Similarly, when $\|\operatorname{FFN}(\operatorname{LN}(\boldsymbol{h}_1^{\ell_0+1/2}))\|$ is small, we obtain $\operatorname{LN}(\boldsymbol{h}_1^{\ell_0+1}) \approx \operatorname{LN}(\boldsymbol{h}_1^{\ell_0})$, which means that $\operatorname{LN}(\boldsymbol{h}_1^{\ell_0}) \approx \cdots \approx \operatorname{LN}(\boldsymbol{h}_1^{\ell_1})$. As for the computation of $\boldsymbol{k}_1^{\ell,h}$ and $\boldsymbol{v}_1^{\ell,h}$ for $\ell_0 \le \ell \le \ell_1$, we have $\boldsymbol{k}_1^{\ell,h} = \boldsymbol{W}_K^{\ell,h} \operatorname{LN}(\boldsymbol{h}_1^\ell) \approx \boldsymbol{W}_K^{\ell,h} \operatorname{LN}(\boldsymbol{h}_1^{\ell_0})$ and $\boldsymbol{v}_1^{\ell,h} = \boldsymbol{W}_V^{\ell,h} \operatorname{LN}(\boldsymbol{h}_1^\ell) \approx \boldsymbol{W}_V^{\ell,h} \operatorname{LN}(\boldsymbol{h}_1^{\ell_0})$, indicating that each $\boldsymbol{k}_1^{\ell,h}$ and $\boldsymbol{v}_1^{\ell,h}$ are nearly generated from the same normalized hidden state $\operatorname{LN}(\boldsymbol{h}_1^{\ell_0})$.

The influence of the first token on the remaining tokens is mediated through its key and value vectors. When massive activations are removed, the hidden states of the first token across layers become more susceptible to perturbations introduced by successive self-attention and FFN blocks through RMSNorm, thereby breaking the approximation described above. As a result, we have $\boldsymbol{k}_1^{\ell,h} \not\approx \boldsymbol{W}_K^{\ell,h} \operatorname{LN}(\boldsymbol{h}_1^{\ell_0})$ and $\boldsymbol{v}_1^{\ell,h} \not\approx \boldsymbol{W}_V^{\ell,h} \operatorname{LN}(\boldsymbol{h}_1^{\ell_0})$. Gu et al. (2024) showed that both $\boldsymbol{k}_1^{\ell,h}, \boldsymbol{h}_1^{\ell,h}$ have a small norm, which can be regarded as an emergent stable state learned by the model. As shown in Figure 6, the norms of $\boldsymbol{k}_1^{\ell,h}$ and $\boldsymbol{v}_1^{\ell,h}$ increase in the absence of massive activations.

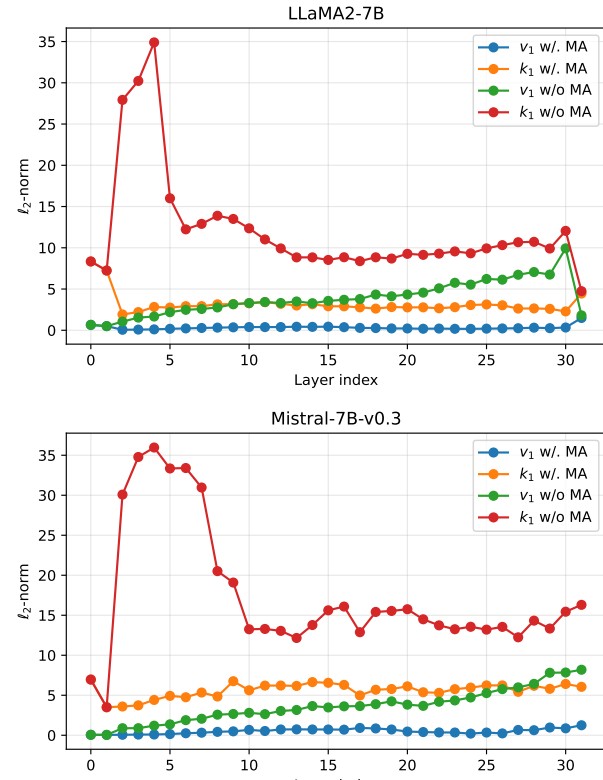

*Figure 6.* Zeroing out the massive activations, KV norm becomes larger and fluctuates across layers.

## 5. Connections to More Phenomena

In this section, we connect previous theoretical and empirical results to three related phenomena: (i) the ineffectiveness of deep layers, (ii) training instability caused by entropy collapse, (iii) low-sensitivity of Transformers.

**Ineffectiveness of deep layers.** Recent works (Gromov et al., 2025; Sandoval-Segura et al., 2025) have observed that removing a large fraction of deeper layers in large language models (LLMs) leads to minimal degradation on downstream tasks. Li et al. (2025) and Sun et al. (2025) attributed this phenomenon to the widely used pre-LayerNorm (pre-LN) modules. Specifically, Theorem 3.3 in (Sun et al., 2025) demonstrates that as hidden states grow larger in deeper layers, updates to the residual stream are largely suppressed by the RMSNorm normalization, resulting in a small gradient norm $\left\|\frac{\partial \boldsymbol{H}^{\ell+1}}{\partial \boldsymbol{H}^\ell}\right\|$. Similarly, our Theorem 4.1 shows that attention sinks with massive activations also lead to a small $\left\|\frac{\partial \boldsymbol{H}^{\ell+1/2}}{\partial \boldsymbol{H}^\ell}\right\|$. However, because massive activations are localized to sink tokens, other tokens with lower-norm hidden states can still be possible to receive meaningful updates from FFNs. As illustrated in Figure 14, FFNs play a more important role in

forming context-aware representations.

**Entropy collapse and training stability.** Entropy collapse refers to the phenomenon where attention weights become overly concentrated on a single token (Zhai et al., 2023; Hong & Lee, 2025), which has been shown to cause training instability and loss spikes. The formation of attention sinks—a specific manifestation of this concentration—also introduces the risk of gradient norm growth and training instability. As demonstrated in Theorem 4.1, the norm $\left\|\frac{\partial \boldsymbol{H}^{\ell+1/2}}{\partial \boldsymbol{H}^{\ell}}\right\|$ increases due to the $\sqrt{T}(1-\varepsilon)$ term induced by the attention sink, potentially leading to loss spikes (Takase et al., 2023). Massive activations serve as a natural remedy to this issue by regularizing $\left\|\frac{\partial \boldsymbol{H}^{\ell+1/2}}{\partial \boldsymbol{H}^{\ell}}\right\|$ via RMSNorm. This mechanism closely parallels the findings in (DeepSeek-AI, 2026), which link the occurrence of loss spikes to the emergence of outlier activations.

**Low sensitivity of Transformer models.** Vasudeva et al. (2024) demonstrated that Transformers exhibit lower sensitivity than MLPs, CNNs, and LSTMs. Specifically, they showed that Transformers learn functions with low sensitivity to small, token-wise input perturbations. Figure 6 in (Vasudeva et al., 2024) illustrates that RoBERTa's sensitivity after layer normalization is uniform across token positions, with the notable exception of the [CLS] token. This aligns with our findings in Figure 2 and strengthens the argument that massive activations are necessary to make sink tokens robust against perturbations. We hypothesize that low-sensitivity bias is possibly one of the inductive biases driving the formation of attention sinks; conversely, attention sinks, coupled with massive activations, represent the primary mechanism underlying this low-sensitivity property of pre-trained Transformers.

## 6. Intervention Analysis

Building on our previous analysis, we show that massive activations can impede the mixing between sink tokens and the remaining tokens, constituting a mechanism that runs in parallel to, rather than causes, the attention sinks. In other words, massive activations are not the root cause of the emergence of attention sinks. As shown in (Sun et al., 2024), setting the massive activations to zero will cause a significant degradation on model performance, but how they will affect the attention structure is unknown. In this section, we extensively investigate how interventions on massive activations can affect attention sinks and answer why modifications to massive activations can nonetheless disrupt the attention sink structure. As a result, we demonstrate that massive activations are not needed to form attention sinks in per-trained models. We report the sink rate $\mathrm{Sink}_1^\varepsilon$ under different interventions on models Llama2-7B (Touvron

et al., 2023) , Llama3-8B, Llama3.1-8B and Llama3.2-3B (Grattafiori et al., 2024). The results are presented in Figure 7 and Table 2.

**Zeroing out and scaling massive activations.** Since the number of massive activations in popular LLMs is less than $5$, we consider zeroing out the top-k massive activations with $k = 1, 2, 3, 4, 5$ when they first appear in the hidden states of [BOS] token, denoted as **Before LN**. As hypothesized in (Gu et al., 2024), after RMSNorm, the input of self-attention layers will be highly sparse, as massive activations will force other positions to zero due to the normalization in RMSNorm. Then we consider zeroing out the top-k largest entries after RMSNorm in the remaining layers, denoted as **After LN**. As shown in Figure 7a, once we zeroing out the massive activations before RMSNorm, $\mathrm{Sink}_1^\epsilon$ decreases sharply to near zero. However, after setting the largest entries after RMSNorm, $\mathrm{Sink}_1^\epsilon$ shows a moderate drop. This demonstrate that the massive activations after RMSNorm may not play an important role in forming attention sinks. We report the index of the top-5 entries before and after LN in Table 4. Note that the top-5 entries before and after RMSNorm may not the same, meaning that we can not simply think that the sparsity induced by RMSNorm will always keep the entries of massive activations.

Since RMSNorm is approximately scale-invariant, i.e., $\mathrm{LN}(\boldsymbol{x}) \approx \mathrm{LN}(\alpha\boldsymbol{x})$ for any $\alpha > 0$, the hidden states of sink tokens do not necessarily need to have large norms to produce the same output. The key difference is that large norms are substantially more insensitive to perturbations. To investigate this phenomenon, we scale the input of RMSNorm at the layer where massive activations first emerge, which we refer to as the **One layer** intervention. We also consider replacing the inputs of all subsequent RMSNorm layers after the emergence of massive activations by the scaled one simultaneously, referred to as the **All layers** intervention. As shown in Figure 7b, due to the algebraic property of RMSNorm, $\mathrm{Sink}_1^\epsilon$ is largely preserved under the All layers intervention, while it decreases sharply under the One layer intervention. This suggests that a large norm of the first token is not required to maintain the same output; however, without consistently preserving a large norm across layers, perturbations introduced by subsequent FFNs and self-attention modules can disrupt the representations of sink tokens.

**Setting value vectors to zero can recover attention sinks.** Zeroing out the massive activations will disrupt the keys and values of the sink token. We use $\boldsymbol{h} = 0$ to denote simply zeroing out massive activations while $\boldsymbol{h} = \boldsymbol{v} = 0$ to denote additionally setting the value vectors corresponding to the sink token to zero. As shown in Table 2, setting massive activations to zero leads to a significant reduction in

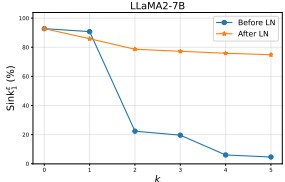 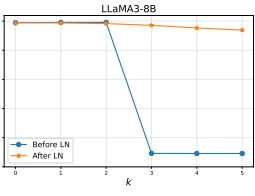

*(a)* Set the top-k with $k = 1, 2, 3, 4, 5$ massive activations to zero.

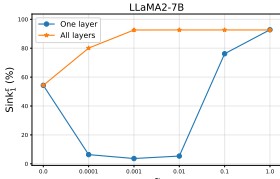 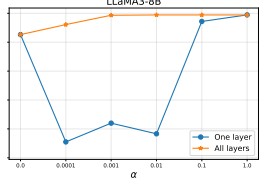

*(b)* Multiplying the hidden state with massive activations by $\alpha = 0, 0.0001, 0.001, 0.01, 0.1, 1.0$.

*Figure 7.* Zeroing out and scaling massive activations.

*Table 2.* Cosine Similarity.

| | Intervention | LLaMA3.1-8B | LLaMA3.2-3B |
|---|---|---|---|
| $\cos(\boldsymbol{k}_1, \boldsymbol{q}_{i\neq1})$ | Original | 0.48 | 0.56 |
| | $\boldsymbol{h} = 0$ | -0.12 | 0.35 |
| | $\boldsymbol{h} = \boldsymbol{v} = 0$ | 0.23 | 0.38 |
| $\cos(\boldsymbol{k}_{i\neq1}, \boldsymbol{q}_{j\neq1})$ | Original | -0.089 | -0.13 |
| | $\boldsymbol{h} = 0$ | 0.25 | 0.16 |
| | $\boldsymbol{h} = \boldsymbol{v} = 0$ | -0.047 | -0.035 |

*Table 3.* Effects of different interventions on the final outputs of Llama2-7B and Llama3-8B. We evaluate the perplexity of the intervened models.

| Model | Intervention | WikiText | C4 | PG-19 |
|---|---|---|---|---|
| LLaMA2-7B | Original | 5.37 | 8.27 | 7.52 |
| | $\boldsymbol{h} = 0$ | 9774.18 | 10667.96 | 7150.94 |
| | $\boldsymbol{h} = \boldsymbol{v} = 0$ | 29.04 | 46.99 | 40.00 |
| LLaMA3-8B | Original | 5.53 | 10.56 | 10.81 |
| | $\boldsymbol{h} = 0$ | 942461.50 | 492007.81 | 304065.96 |
| | $\boldsymbol{h} = \boldsymbol{v} = 0$ | 452.87 | 424.11 | 330.29 |

$\text{Sink}_1^\varepsilon$ for $\varepsilon = 0.3, 0.5, 0.8$. However, zeroing out the value vector corresponding to the sink token restores the attention sink. As shown in (Gu et al., 2024), achieving a large $\boldsymbol{q}_t^{\ell,h} \boldsymbol{k}_1^{\ell,h\top}$ primarily relies on a comparatively large cosine similarity $\cos\left(\boldsymbol{q}_t^{\ell,h}, \boldsymbol{k}_1^{\ell,h}\right)$, rather than on the magnitude of $\|\boldsymbol{k}_1^{\ell,h}\|$. In contrast, the cosine similarities $\cos\left(\boldsymbol{q}_t^{\ell,h}, \boldsymbol{k}_{j\neq1}^{\ell,h}\right)$ are typically negative (see a similar observation in Figure 5 of (Sun et al., 2024)). As shown in Table 2, an improper value vector for the sink token disrupts this negative cosine-similarity structure among non-sink tokens, thereby breaking the formation of attention sink. Consequently, setting the value vector of the sink token to zero restores the attention sink. This result further confirms that massive activations are not necessary for the formation of attention sink in pre-trained models.

We find that under the intervention $\boldsymbol{h} = \boldsymbol{v} = 0$, the attention sink becomes even stronger when $\varepsilon$ is large. Although the cosine similarity $\cos(\boldsymbol{k}_1, \boldsymbol{q}_{i\neq1})$ slightly decreases in Table 2, Figure 6 shows that the norm of $\boldsymbol{k}_1$ increases, resulting in a larger inner product $\langle \boldsymbol{k}_1, \boldsymbol{q}_{i\neq1} \rangle$.

*Table 1.* Intervention Analysis.

| Model | Intervention | $\text{Sink}_1^{0.3}$ | $\text{Sink}_1^{0.5}$ | $\text{Sink}_1^{0.8}$ |
|---|---|---|---|---|
| LLaMA3.1-8B | Original | 98.91 | 88.95 | 45.70 |
| | $\boldsymbol{h} = 0$ | 9.19 | 8.18 | 5.69 |
| | $\boldsymbol{h} = \boldsymbol{v} = 0$ | 96.60 | 87.94 | 60.38 |
| LLaMA3.2-3B | Original | 98.91 | 88.79 | 43.62 |
| | $\boldsymbol{h} = 0$ | 34.95 | 19.50 | 10.80 |
| | $\boldsymbol{h} = \boldsymbol{v} = 0$ | 97.12 | 93.71 | 75.41 |

**The influence on model performance.** We evaluate the perplexity on WikiText (Merity et al., 2016), C4 (Raffel

et al., 2020), and PG-19 (Rae et al., 2019) under different intervention strategies. Setting massive activations to zero leads to a sharp increase in perplexity, reproducing the results in (Sun et al., 2024). However, when the corresponding value vectors are set to zero simultaneously after the layer where massive activations first emerge, model performance is substantially recovered, although it remains noticeable worse than original baseline. This implies that the recovered attention sink is not functionally equivalent to the original one. In Figures 11 and 12, we visualize the attention patterns under these interventions. We find that while the overall patterns remain similar, they differ substantially in the interactions among non-sink tokens.

# 7. Conclusion

In this work, we provide a mechanistic understanding of the relationship between massive activations and attention sink in Transformer models. Through theoretical analysis and interventions, we show that massive activations are not necessary for attention sinks but act to stabilize self-attention by suppressing mixing between sink and non-sink tokens. We explain why sink tokens tend to develop large norms under RMSNorm and how design choices such as KV biases, gating, and alternative attention formulations can mitigate massive activations while largely preserving attention sinks. We further show that improper manipulation of massive activations can disrupt attention sinks and degrade performance, whereas removing the sink token's value vectors can restore them. Overall, our results disentangle the roles of massive activations and attention sinks, offering insights into model stability. A limitation of our study is that the formation dynamics of massive activations and attention sinks during training and their long-term impact on training stability remain underexplored.

## Impact statement

This work advances the mechanistic understanding of Transformer models by clarifying the roles of attention sink and massive activations in regulating token mixing and representation stability. By disentangling these two phenomena, our findings may inform the design of more stable and interpretable architectures, as well as safer and more principled intervention strategies for model modification, compression, quantization, and optimization. While this study does not introduce new model capabilities or applications, improved understanding of internal mechanisms can contribute indirectly to the development of more reliable large language models. We do not anticipate direct negative societal impacts from this work; however, as with all research on large-scale models, downstream applications should continue to be developed and deployed with appropriate care and oversight.

## Acknowledgements

Yuanyuan Lin's research was partially supported by the Hong Kong Research Grants Council (No. 14304523) and The Hong Kong Jockey Club Equine Welfare Research Foundation, Direct Grants for Research, The Chinese University of Hong Kong.

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

# A. Additional Experiments

## A.1. Reproduced Results

For completeness and clearness, we reproduce some classical results about massive activations in existing literature (Sun et al., 2024; Gu et al., 2024). In Figure 8, we plot the massive activations in [BOS] token of Llama2-7B and Llama3-8B, respectively.

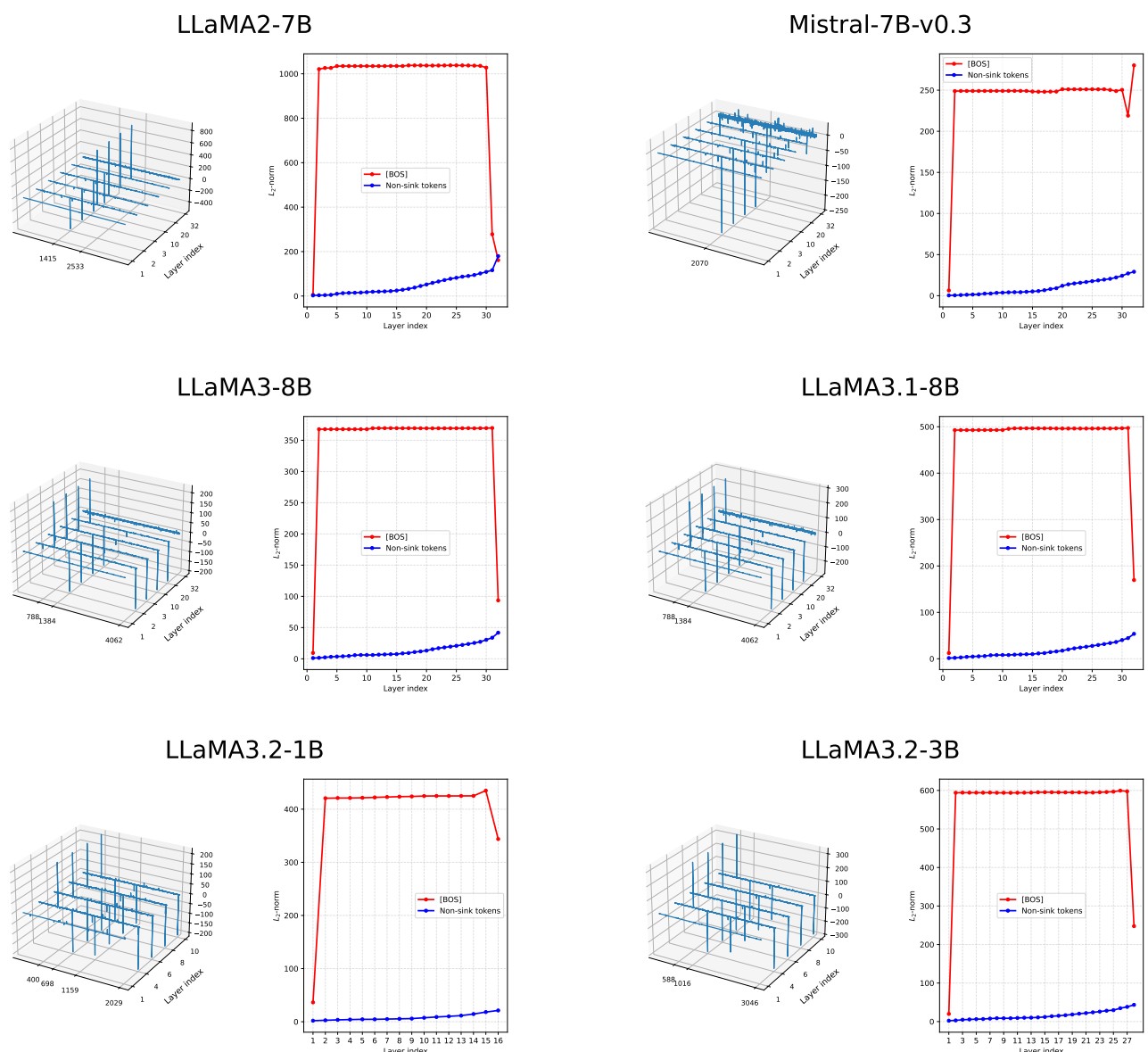

*Figure 8.* The visualization of massive activations in various pre-trained LLMs.

## A.2. The Effect of Different Types of Intervention on Massive Activations

In Section 6, we conduct several kinds of intervention to the sink token's hidden states and it is unclear their influence on their massive activations. As shown in Figure 9, massive activations disappear under four kinds of intervention: **(a): swapping massive activations to random position; (b): setting the entire hidden state to zero when the massive activations first appear; (c): setting two massive activations to zero when thry first emerge; (d): Scaling the hidden state by 0.1 when massive activations first appear**. In Figure 10, we plot the norm of [BOS] under different interventions.

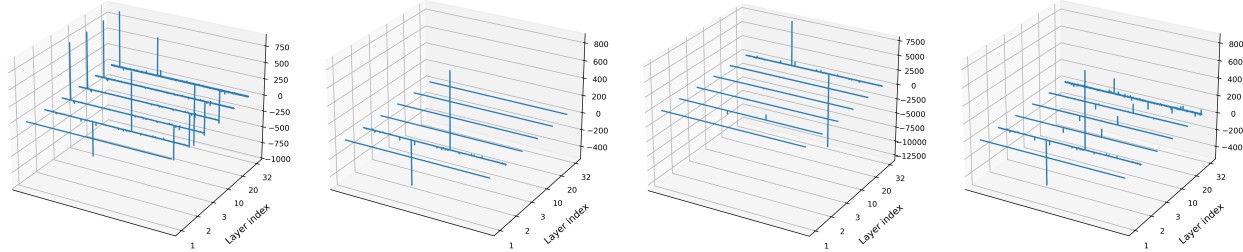

*(a)* Swap the massive activations to random positions. *(b)* Set the entire hidden state to zero. *(c)* Set two massive activations to zero *(d)* multiply the hidden state by 0.1

*Figure 9.* **Output of each layer under four different Operations done to [BOS]'s hidden state with massive activations.** In Llama2-7B, the massive activations emerge in the output of the second layer's FFN. We consider changing the input of the third layer by four ways: (a) Change the position of massive activations to a random one while keep their magnitude fixed. (b) Set the entire hidden state to zero, that is, let $\boldsymbol{h}_1^2 = \boldsymbol{0}$. (c) Set the two massive activations located in the 1415-th and 2533-th elements to zero, that is, let $(\boldsymbol{h}_1^2)_{1415} = (\boldsymbol{h}_1^2)_{2533} = 0$. (d) Multiply $\boldsymbol{h}_1^2$ by a scaling factor $\alpha = 0.1$.

## A.3. The Indices of Massive Activations Before and After LN

Gu et al. (2024) assumed that massive activations are largely preserved after layer normalization (LN), while other activations are suppressed toward zero, and that these preserved entries contribute significantly to the formation of attention sink. We've already shown that these entries have nearly nothing to do with attention sink. For completeness, we examine the indices of massive activations before and after LN and find that they do not always coincide. In fact, under RMSNorm or layer normalization, the top-k activations are not necessarily preserved, as the per-layer scaling factors in each normalization module can reorder activation magnitudes.

*Table 4.* The indices of top5 activations in the hidden states of layer 8 in LLaMA3.2-1B and 3B, and hidden states of layer 16 of remaining models.

| Model | Indices |
|---|---|
| LLaMA2-7B | Befor RMSNorm: [2533, 1415, 1076, 1512, 2298] |
| | After RMSNorm: [2533, 1076, 2789, 1415, 3431] |
| LLaMA3-8B | Before RMSNorm: [788, 1384, 4062, 2352, 290] |
| | After RMSNorm: [ 788, 4062, 1384, 290, 2352] |
| Mistral-7B-v0.3 | Before RMSNorm: [2070, 3398, 1935, 3701, 3071] |
| | After RMSNorm: [2070, 1935, 3701, 3398, 1437] |
| LLaMA3.1-8B | Before RMSNorm: [788, 1384, 4062, 2352, 290] |
| | After RMSNorm: [788, 4062, 1384, 290, 2352] |
| LLaMA3.2-1B | Before RMSNorm: [ 400, 698, 2029, 1159, 2023] |
| | After RMSNorm: [400, 698, 2029, 1159, 1107] |
| LLaMA3.2-3B | Before RMSNorm: [588, 1016, 3046, 1731, 1659] |
| | After RMSNorm: [588, 1016, 3046, 1731, 1659] |

## A.4. Recovered Attention Sink

In Figure 11 and 12, we provide the visualization of attention patterns from the 4-th layer and the 4-th head under different interventions.

## Llama2-7B

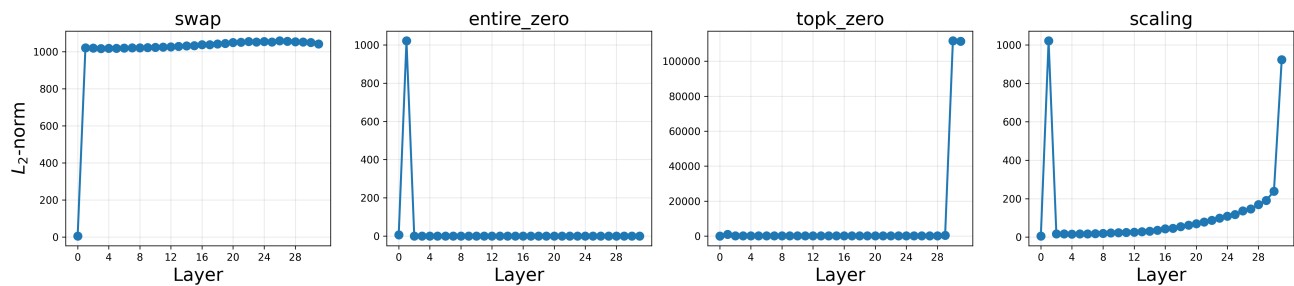

*(a)* Norm of [BOS] in LLaMA2-7B under different interventions.

## Llama3-8B

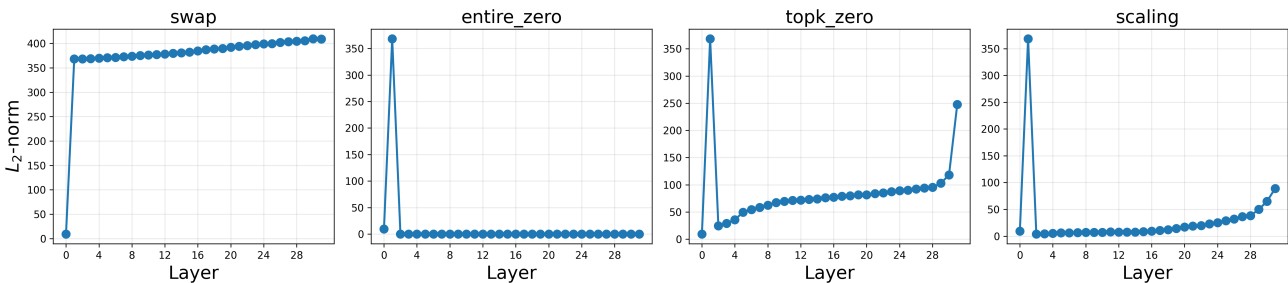

*(b)* Norm of `[BOS]` in LLaMA3-8B under different interventions.

## Mistral-7B

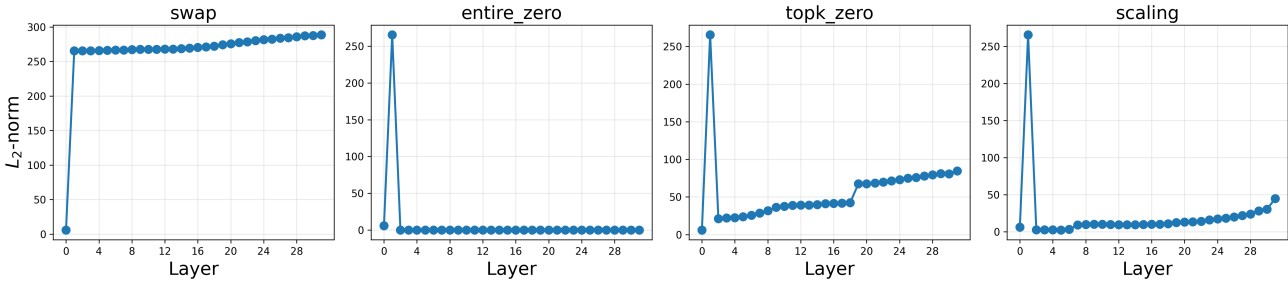

*(c)* Norm of `[BOS]` in Mistrial-7B under different interventions.

*Figure 10.* The effects of different interventions on massive activations.

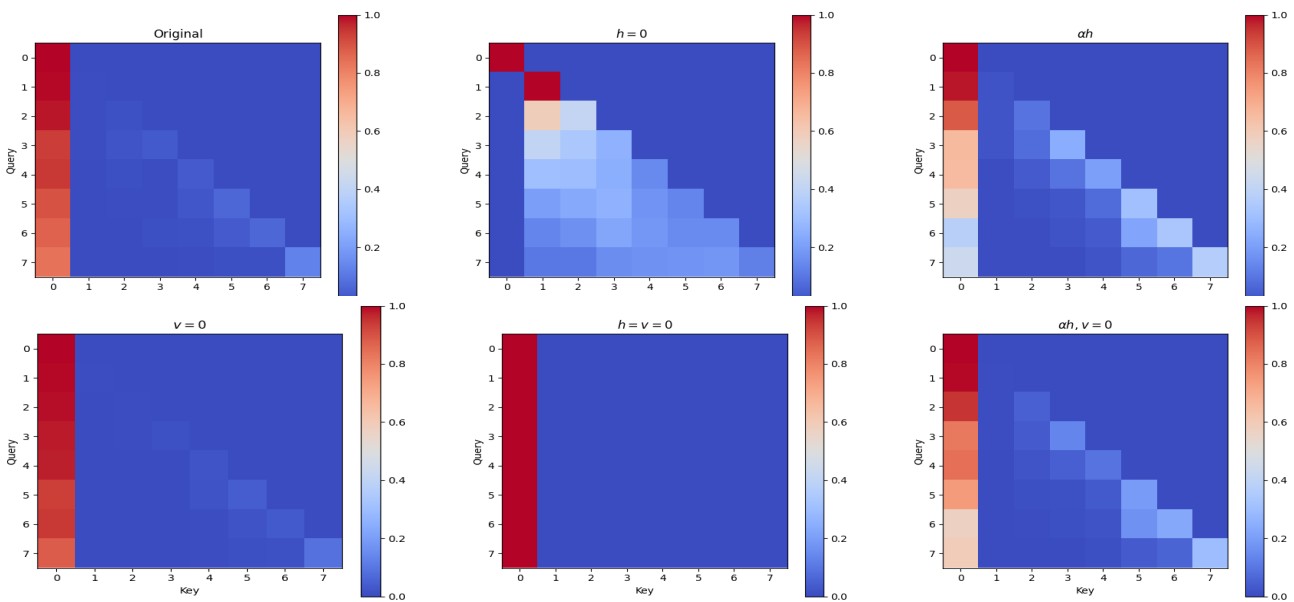

Figure 11. Attention patterns of LLaMA2-7B under different interventions.

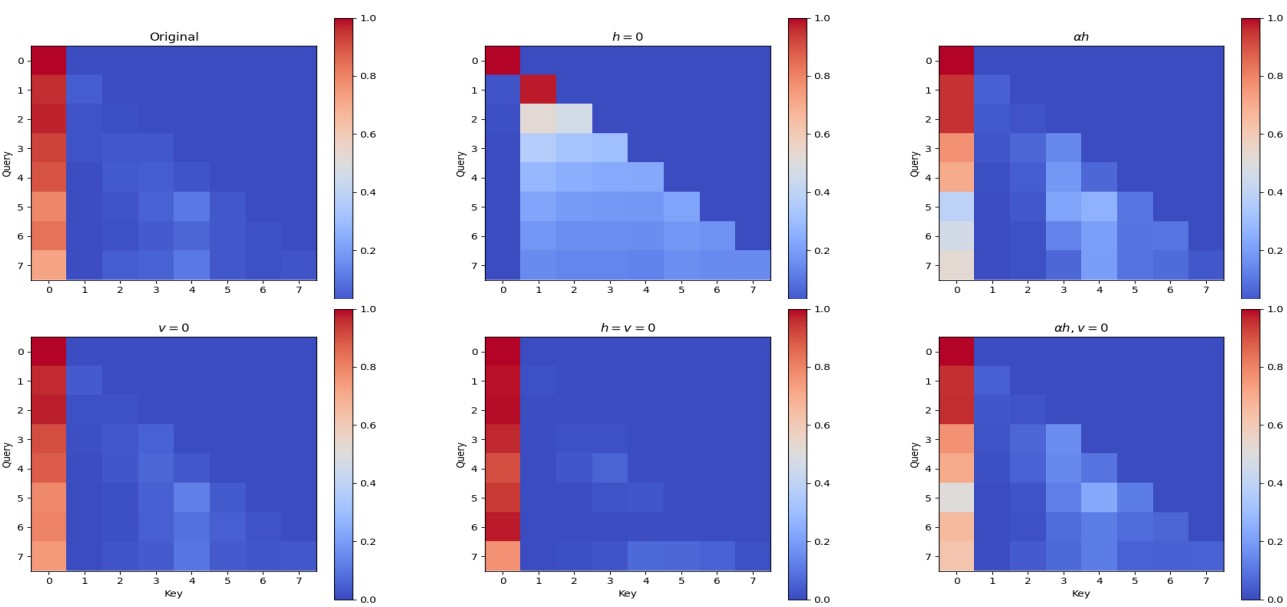

Figure 12. Attention patterns of LLaMA3-8B under different interventions.

# B. Proofs

In this section, we provede the proof of Theorem 4.1. We will first introduce the notation for gradient computation, as well as some basic properties about matrix analysis following (Noci et al., 2022).

## B.1. Prelieminaries

To compute the Jacobian matrix of matritrx function, we use the conventional method of matrix vectorization. Let $\boldsymbol{F} = (f_{ij}) : \mathbb{R}^{m \times n} \to \mathbb{R}^{p \times q}$ be a matrix function, then we have

$$\frac{\partial \boldsymbol{F}(\boldsymbol{X})}{\partial \boldsymbol{X}} := \frac{\partial \operatorname{vec}(\boldsymbol{F}(\boldsymbol{X}))}{\partial \operatorname{vec}(\boldsymbol{X})^\top} \in \mathbb{R}^{pq \times mn},$$

where $\operatorname{vec}(\cdot)$ is the column vectorization. Specifically, we have

$$\frac{\partial \operatorname{vec}(\boldsymbol{F}(\boldsymbol{X}))}{\partial \operatorname{vec}(\boldsymbol{X})^\top} = \begin{bmatrix} \frac{\partial f_{11}}{\partial x_{11}} & \cdots & \frac{\partial f_{11}}{\partial x_{m1}} & \cdots & \frac{\partial f_{11}}{\partial x_{1n}} & \cdots & \frac{\partial f_{11}}{\partial x_{mn}} \\ \vdots & \vdots & \vdots & \vdots & \vdots & \vdots & \vdots \\ \frac{\partial f_{p1}}{\partial x_{11}} & \cdots & \frac{\partial f_{p1}}{\partial x_{m1}} & \cdots & \frac{\partial f_{p1}}{\partial x_{1n}} & \cdots & \frac{\partial f_{11}}{\partial x_{mn}} \\ \vdots & \vdots & \vdots & \vdots & \vdots & \vdots & \vdots \\ \frac{\partial f_{1q}}{\partial x_{11}} & \cdots & \frac{\partial f_{1q}}{\partial x_{m1}} & \cdots & \frac{\partial f_{1q}}{\partial x_{1n}} & \cdots & \frac{\partial f_{1q}}{\partial x_{mn}} \\ \vdots & \vdots & \vdots & \vdots & \vdots & \vdots & \vdots \\ \frac{\partial f_{pq}}{\partial x_{11}} & \cdots & \frac{\partial f_{pq}}{\partial x_{m1}} & \cdots & \frac{\partial f_{pq}}{\partial x_{1n}} & \cdots & \frac{\partial f_{pq}}{\partial x_{mn}} \end{bmatrix}.$$

Let $\otimes$ denote the kronecker product with $A \otimes \boldsymbol{B} = \begin{bmatrix} a_{11}\boldsymbol{B} & \cdots & a_{1n}\boldsymbol{B} \\ \vdots & \ddots & \vdots \\ a_{m1}\boldsymbol{B} & \cdots & a_{mn}\boldsymbol{B} \end{bmatrix}$. In the following, we list some basics of matrix differentiation. For any $\boldsymbol{X} \in \mathbb{R}^{m \times n}$, we have

$$\frac{\partial \boldsymbol{X}}{\partial \boldsymbol{X}} = \boldsymbol{I}_n \otimes \boldsymbol{I}_m = \boldsymbol{I}_{mn}, \quad \frac{\partial \boldsymbol{A}\boldsymbol{X}\boldsymbol{B}}{\partial \boldsymbol{X}} = \boldsymbol{B} \otimes \boldsymbol{A}^\top.$$

We also use some properties of Kronecker product. Given $\boldsymbol{A} \in \mathbb{R}^{m \times n}$, $\boldsymbol{B} \in \mathbb{R}^{p \times q}$, and $\boldsymbol{C} \in \mathbb{R}^{n \times r}$, $\boldsymbol{D} \in \mathbb{R}^{q \times s}$, then the following holds:

$$\operatorname{tr}(\boldsymbol{A} \otimes \boldsymbol{B}) = \operatorname{tr}(\boldsymbol{A})\operatorname{tr}(\boldsymbol{B}),$$

and

$$(\boldsymbol{A} \otimes \boldsymbol{B})(\boldsymbol{C} \otimes \boldsymbol{D}) = (\boldsymbol{A}\boldsymbol{C}) \otimes (\boldsymbol{B}\boldsymbol{D}).$$

## B.2. Discussion on the Assumptions in Theorem 4.1

Recall that in Theorem 4.1, we assume that for the $(\ell + 1)$-th self-attention layer, the attention probability matrix satisfies $\boldsymbol{A}_{1,i}^{\ell+1,h} = 1 - \varepsilon_i$ for any $i \in \{2, \cdots, T\}$ and $h \in [H]$, where $\epsilon \le \varepsilon_i \le \varepsilon'$. First, we only require that $\varepsilon_i \in (0, 1)$, it does not have to be very small. However, as $\varepsilon$ approaches zero, the mixing among non-sink tokens is further suppressed. This assumption is consistent with practical observations, where $\varepsilon$ is not necessarily very small, particularly when the sequence length $T$ is large. The key point is that $\frac{\varepsilon}{\sqrt{T-1}}$ is very small ($\varepsilon < 1$), which roughly represents the attention score non-sink tokens can receive.

Second, we assume that each head in the $(\ell + 1)$-th layer exhibits attention sink phenomenon. While in practice, this may not always hold, it is reasonable for our analysis because we focus on the relationship between massive activations and attention sinks; ignoring heads that mix tokens allows us to isolate this effect. Equivalently, we can assume a subset of these heads exhibiting attention sinks, and only consider their corresponding outputs.

## B.3. Proof of Theorem 4.1

*Proof of Theorem 4.1.* Since multi-head self-attention is nothing but a linear combination of single head attention, in this proof we start with single head self-attention mechanism for notational simplicity and absorb $\boldsymbol{W}_O$ into $\boldsymbol{W}_V$.

For any $\boldsymbol{H} = [\boldsymbol{h}_1, \cdots, \boldsymbol{h}_T] \in \mathbb{R}^{d \times T}$, let $\boldsymbol{X} = \mathrm{LN}(\boldsymbol{H}) = [\boldsymbol{x}_1, \cdots, \boldsymbol{x}_T] \in \mathbb{R}^{d \times T}$, $\boldsymbol{T} = \boldsymbol{W}_V \boldsymbol{X}$, $\boldsymbol{A} = \mathrm{Softmax}\left[\frac{1}{\sqrt{d}}(\boldsymbol{W}_K \boldsymbol{X})^\top(\boldsymbol{W}_Q \boldsymbol{X})\right]$, $\boldsymbol{M} = \frac{1}{\sqrt{d}}\boldsymbol{X}^\top \boldsymbol{W}_K^\top \boldsymbol{W}_Q \boldsymbol{X}$, $\boldsymbol{S} = \boldsymbol{T}\boldsymbol{A} = \mathrm{MHSA}(\boldsymbol{X}) = \boldsymbol{W}_V \boldsymbol{X}\,\mathrm{Softmax}\left[\frac{1}{\sqrt{d}}(\boldsymbol{W}_K \boldsymbol{X})^\top(\boldsymbol{W}_Q \boldsymbol{X})\right]$. In the following, we try to derive $\frac{\partial(\boldsymbol{S}+\boldsymbol{H})}{\partial \boldsymbol{H}}$. We basically follow the framework in (Noci et al., 2022) which processes the contributions from $\boldsymbol{A}$ and $\boldsymbol{T}$ respectively.

By the chain rule, we have

$$\frac{\partial(\boldsymbol{S}+\boldsymbol{H})}{\partial \boldsymbol{H}} = \frac{\partial \boldsymbol{S}}{\partial \boldsymbol{X}}\frac{\partial \boldsymbol{X}}{\partial \boldsymbol{H}} + \frac{\partial \boldsymbol{H}}{\partial \boldsymbol{H}},$$

where $\frac{\partial \boldsymbol{X}}{\partial \boldsymbol{H}}$ can be derived directly from Lemma B.1, so we mainly focus on $\frac{\partial \boldsymbol{S}}{\partial \boldsymbol{X}}$. Similarly, we have

$$\begin{aligned}
\frac{\partial \boldsymbol{S}}{\partial \boldsymbol{X}} &= \frac{\partial \boldsymbol{S}}{\partial \boldsymbol{A}}\frac{\partial \boldsymbol{A}}{\partial \boldsymbol{X}} + \frac{\partial \boldsymbol{S}}{\partial \boldsymbol{T}}\frac{\partial \boldsymbol{T}}{\partial \boldsymbol{X}} \\
&= \frac{\partial \boldsymbol{S}}{\partial \boldsymbol{A}}\frac{\partial \boldsymbol{A}}{\partial \boldsymbol{M}}\frac{\partial \boldsymbol{M}}{\partial \boldsymbol{X}} + \frac{\partial \boldsymbol{S}}{\partial \boldsymbol{T}}\frac{\partial \boldsymbol{T}}{\partial \boldsymbol{X}} \\
&= (\boldsymbol{I}_n \otimes \boldsymbol{W}_V \boldsymbol{X})\frac{\partial \boldsymbol{A}}{\partial \boldsymbol{M}}\frac{\partial \boldsymbol{M}}{\partial \boldsymbol{X}} + (\boldsymbol{A}^\top \otimes \boldsymbol{I}_d)(\boldsymbol{I}_n \otimes \boldsymbol{W}_V) \\
&= (\boldsymbol{I}_n \otimes \boldsymbol{W}_V \boldsymbol{X})\frac{\partial \boldsymbol{A}}{\partial \boldsymbol{M}}\frac{\partial \boldsymbol{M}}{\partial \boldsymbol{X}} + (\boldsymbol{A}^\top \otimes \boldsymbol{W}_V).
\end{aligned} \tag{3}$$

As for $\frac{\partial \boldsymbol{M}}{\partial \boldsymbol{X}}$, we can derive

$$\mathrm{d}\boldsymbol{M} = \frac{1}{\sqrt{d}}\,\mathrm{d}(\boldsymbol{X}^\top)\boldsymbol{W}_K^\top \boldsymbol{W}_Q \boldsymbol{X} + \frac{1}{\sqrt{d}}\boldsymbol{X}^\top \boldsymbol{W}_K^\top \boldsymbol{W}_Q\,\mathrm{d}(\boldsymbol{X}).$$

vectorizing both sides:

$$\begin{aligned}
\mathrm{d}\,\mathrm{vec}(\boldsymbol{M}) &= \frac{1}{\sqrt{d}}(\boldsymbol{X}^\top \boldsymbol{W}_Q^\top \boldsymbol{W}_K \otimes \boldsymbol{I}_n)\,\mathrm{d}\,\mathrm{vec}(\boldsymbol{X}^\top) + \frac{1}{\sqrt{d}}(\boldsymbol{I}_n \otimes \boldsymbol{X}^\top \boldsymbol{W}_K^\top \boldsymbol{W}_Q)\,\mathrm{d}\,\mathrm{vec}(\boldsymbol{X}) \\
&= \frac{1}{\sqrt{d}}(\boldsymbol{X}^\top \boldsymbol{W}_Q^\top \boldsymbol{W}_K \otimes \boldsymbol{I}_n)\,\mathrm{d}\boldsymbol{K}_{dn}\,\mathrm{d}\,\mathrm{vec}(\boldsymbol{X}) + \frac{1}{\sqrt{d}}(\boldsymbol{I}_n \otimes \boldsymbol{X}^\top \boldsymbol{W}_K^\top \boldsymbol{W}_Q)\,\mathrm{d}\,\mathrm{vec}(\boldsymbol{X}),
\end{aligned}$$

where $\boldsymbol{K}_{dn}$ is the commutative matrix. Then we have

$$\frac{\partial \boldsymbol{M}}{\partial \boldsymbol{X}} = \frac{1}{\sqrt{d}}(\boldsymbol{X}^\top \boldsymbol{W}_Q^\top \boldsymbol{W}_K \otimes \boldsymbol{I}_n)\boldsymbol{K}_{dn} + \frac{1}{\sqrt{d}}(\boldsymbol{I}_n \otimes \boldsymbol{X}^\top \boldsymbol{W}_K^\top \boldsymbol{W}_Q).$$

Plug this into Eq.(3), we have

$$\frac{\partial \boldsymbol{S}}{\partial \boldsymbol{X}} = \frac{1}{\sqrt{d}}(\boldsymbol{I}_n \otimes \boldsymbol{W}_V \boldsymbol{X})\frac{\partial \boldsymbol{A}}{\partial \boldsymbol{M}}\left((\boldsymbol{X}^\top \boldsymbol{W}_Q^\top \boldsymbol{W}_K \otimes \boldsymbol{I}_n)\boldsymbol{K}_{dn} + (\boldsymbol{I}_n \otimes \boldsymbol{X}^\top \boldsymbol{W}_K^\top \boldsymbol{W}_Q)\right) + \boldsymbol{A}^\top \otimes \boldsymbol{W}_V.$$

It is clear that $\frac{\partial \boldsymbol{S}}{\partial \boldsymbol{X}} = \begin{pmatrix} \boldsymbol{J}_{11} & \cdots & \boldsymbol{J}_{1n} \\ \vdots & \vdots & \vdots \\ \boldsymbol{J}_{n1} & \cdots & \boldsymbol{J}_{nn} \end{pmatrix}$, where $\boldsymbol{J}_{ij} = \frac{\partial \boldsymbol{S}_{:,i}}{\partial \boldsymbol{X}_{:,j}}$. Note that we do not utilize causal mask in the computation

of each self-attention module to include more situations. To compute each $\boldsymbol{J}_{:,j}$ and $\boldsymbol{J}_{ij}$, we have the following fact

$$
\begin{aligned}
\boldsymbol{J}_{:,j} &= \frac{\partial \boldsymbol{S}}{\partial \boldsymbol{X}_{:,j}} = \frac{\partial \boldsymbol{S}}{\partial \boldsymbol{X}} (\boldsymbol{e}_j \otimes \boldsymbol{I}_d) \\
&= \frac{1}{\sqrt{d}} (\boldsymbol{I}_n \otimes \boldsymbol{W}_V \boldsymbol{X}) \frac{\partial \boldsymbol{A}}{\partial \boldsymbol{M}} \left( (\boldsymbol{X}^\top \boldsymbol{W}_Q^\top \boldsymbol{W}_K \otimes \boldsymbol{I}_n) \boldsymbol{K}_{dn} + (\boldsymbol{I}_n \otimes \boldsymbol{X}^\top \boldsymbol{W}_K^\top \boldsymbol{W}_Q) \right) (\boldsymbol{e}_j \otimes \boldsymbol{I}_d) \\
&\quad + (\boldsymbol{A}^\top \otimes \boldsymbol{W}_V)(\boldsymbol{e}_j \otimes \boldsymbol{I}_d) \\
&= \frac{1}{\sqrt{d}} \operatorname{diag}(\boldsymbol{W}_V \boldsymbol{X} \frac{\partial \boldsymbol{A}_{:,1}}{\partial \boldsymbol{M}_{:,1}}, \cdots, \boldsymbol{W}_V \boldsymbol{X} \frac{\partial \boldsymbol{A}_{:,n}}{\partial \boldsymbol{M}_{:,n}}) \left( \boldsymbol{X}^\top \boldsymbol{W}_Q^\top \boldsymbol{W}_K \otimes \boldsymbol{e}_j + \boldsymbol{e}_j \otimes \boldsymbol{X}^\top \boldsymbol{W}_K^\top \boldsymbol{W}_Q \right) + \boldsymbol{A}_{j,:} \otimes \boldsymbol{W}_V,
\end{aligned}
$$

and

$$
\begin{aligned}
\boldsymbol{J}_{ij} &= (\boldsymbol{e}_i^\top \otimes \boldsymbol{I}_d) \frac{\partial \boldsymbol{S}}{\partial \boldsymbol{X}} (\boldsymbol{e}_j \otimes \boldsymbol{I}_d) \\
&= \frac{1}{\sqrt{d}} (\boldsymbol{e}_i^\top \otimes \boldsymbol{I}_d)(\boldsymbol{I}_n \otimes \boldsymbol{W}_V \boldsymbol{X}) \frac{\partial \boldsymbol{A}}{\partial \boldsymbol{M}} \left( (\boldsymbol{X}^\top \boldsymbol{W}_Q^\top \boldsymbol{W}_K \otimes \boldsymbol{I}_n) \boldsymbol{K}_{dn} + (\boldsymbol{I}_n \otimes \boldsymbol{X}^\top \boldsymbol{W}_K^\top \boldsymbol{W}_Q) \right) (\boldsymbol{e}_j \otimes \boldsymbol{I}_d) \\
&\quad + (\boldsymbol{e}_i^\top \otimes \boldsymbol{I}_d)(\boldsymbol{A}^\top \otimes \boldsymbol{W}_V)(\boldsymbol{e}_j \otimes \boldsymbol{I}_d) \\
&= \frac{1}{\sqrt{d}} \boldsymbol{e}_i^\top \otimes \boldsymbol{W}_V \boldsymbol{X} \frac{\partial \boldsymbol{A}_{:,i}}{\partial \boldsymbol{M}_{:,i}} \left( \boldsymbol{X}^\top \boldsymbol{W}_Q^\top \boldsymbol{W}_K \otimes \boldsymbol{e}_j + \boldsymbol{e}_j \otimes \boldsymbol{X}^\top \boldsymbol{W}_K^\top \boldsymbol{W}_Q \right) + \boldsymbol{W}_V \cdot \boldsymbol{A}_{i,j}^\top \\
&= \frac{1}{\sqrt{d}} \boldsymbol{W}_V \boldsymbol{X} \frac{\partial \boldsymbol{A}_{:,i}}{\partial \boldsymbol{M}_{:,i}} \boldsymbol{e}_j \boldsymbol{e}_i^\top \boldsymbol{X}^\top \boldsymbol{W}_Q^\top \boldsymbol{W}_K + \boldsymbol{e}_i^\top \boldsymbol{e}_j \boldsymbol{W}_V \boldsymbol{X} \frac{\partial \boldsymbol{A}_{:,i}}{\partial \boldsymbol{M}_{:,i}} \boldsymbol{X}^\top \boldsymbol{W}_K^\top \boldsymbol{W}_Q + \boldsymbol{W}_V \cdot \boldsymbol{A}_{i,j}^\top \\
&= \frac{1}{\sqrt{d}} \boldsymbol{W}_V \boldsymbol{X} \frac{\partial \boldsymbol{A}_{:,i}}{\partial \boldsymbol{M}_{:,i}} \boldsymbol{E}_{ji} \boldsymbol{X}^\top \boldsymbol{W}_Q^\top \boldsymbol{W}_K + \delta_{ij} \boldsymbol{W}_V \boldsymbol{X} \frac{\partial \boldsymbol{A}_{:,i}}{\partial \boldsymbol{M}_{:,i}} \boldsymbol{X}^\top \boldsymbol{W}_K^\top \boldsymbol{W}_Q + \boldsymbol{W}_V \cdot \boldsymbol{A}_{i,j}^\top.
\end{aligned}
$$

Then, we know that

$$
\frac{\partial \operatorname{MHSA}(\operatorname{LN}(\boldsymbol{H}))}{\partial \operatorname{LN}(\boldsymbol{H})} \frac{\partial \operatorname{LN}(\boldsymbol{H})}{\partial \boldsymbol{H}} + \frac{\partial \boldsymbol{H}}{\partial \boldsymbol{H}} = \begin{pmatrix} \boldsymbol{J}_{11} \frac{\partial \operatorname{LN}(\boldsymbol{h}_1)}{\partial \boldsymbol{h}_1} + \boldsymbol{I}_d & \cdots & \boldsymbol{J}_{1n} \frac{\partial \operatorname{LN}(\boldsymbol{h}_n)}{\partial \boldsymbol{h}_n} + \boldsymbol{I}_d \\ \vdots & \vdots & \vdots \\ \boldsymbol{J}_{n1} \frac{\partial \operatorname{LN}(\boldsymbol{h}_1)}{\partial \boldsymbol{h}_1} + \boldsymbol{I}_d & \cdots & \boldsymbol{J}_{nn} \frac{\partial \operatorname{LN}(\boldsymbol{h}_n)}{\partial \boldsymbol{h}_n} + \boldsymbol{I}_d \end{pmatrix}. \tag{4}
$$

By the definition of $\epsilon$ and $\epsilon'$, we have $\epsilon \leq \min_{i=1,\cdots,T} \epsilon_i$ and $\epsilon' \geq \max_{i=1,\cdots,T} \epsilon_i$. To bound each $\|\boldsymbol{J}_{:,j}\|$, we follow the following decomposition

$$
\begin{aligned}
\|\boldsymbol{J}_{:,j}\| &= \left\| \frac{1}{\sqrt{d}} \operatorname{diag}(\boldsymbol{W}_V \boldsymbol{X} \frac{\partial \boldsymbol{A}_{:,1}}{\partial \boldsymbol{M}_{:,1}}, \cdots, \boldsymbol{W}_V \boldsymbol{X} \frac{\partial \boldsymbol{A}_{:,n}}{\partial \boldsymbol{M}_{:,n}}) \left( \boldsymbol{X}^\top \boldsymbol{W}_Q^\top \boldsymbol{W}_K \otimes \boldsymbol{e}_j + \boldsymbol{e}_j \otimes \boldsymbol{X}^\top \boldsymbol{W}_K^\top \boldsymbol{W}_Q \right) + \boldsymbol{A}_{j,:} \otimes \boldsymbol{W}_V \right\| \\
&\leq \underbrace{\left\| \frac{1}{\sqrt{d}} \operatorname{diag}(\boldsymbol{W}_V \boldsymbol{X} \frac{\partial \boldsymbol{A}_{:,1}}{\partial \boldsymbol{M}_{:,1}}, \cdots, \boldsymbol{W}_V \boldsymbol{X} \frac{\partial \boldsymbol{A}_{:,n}}{\partial \boldsymbol{M}_{:,n}}) \left( \boldsymbol{X}^\top \boldsymbol{W}_Q^\top \boldsymbol{W}_K \otimes \boldsymbol{e}_j \right) \right\|}_{\text{(I)}} \\
&\quad + \underbrace{\left\| \frac{1}{\sqrt{d}} \operatorname{diag}(\boldsymbol{W}_V \boldsymbol{X} \frac{\partial \boldsymbol{A}_{:,1}}{\partial \boldsymbol{M}_{:,1}}, \cdots, \boldsymbol{W}_V \boldsymbol{X} \frac{\partial \boldsymbol{A}_{:,n}}{\partial \boldsymbol{M}_{:,n}}) \left( \boldsymbol{e}_j \otimes \boldsymbol{X}^\top \boldsymbol{W}_K^\top \boldsymbol{W}_Q \right) \right\|}_{\text{(II)}} + \underbrace{\| \boldsymbol{A}_{j,:} \otimes \boldsymbol{W}_V \|}_{\text{(III)}}.
\end{aligned}
$$

In the following, we bound (I), (II) and (III) separately. Let's begin with (I).

$$
\begin{aligned}
\text{(I)} : &\left\| \frac{1}{\sqrt{d}} \operatorname{diag}(\boldsymbol{W}_V \boldsymbol{X} \frac{\partial \boldsymbol{A}_{:,1}}{\partial \boldsymbol{M}_{:,1}}, \cdots, \boldsymbol{W}_V \boldsymbol{X} \frac{\partial \boldsymbol{A}_{:,n}}{\partial \boldsymbol{M}_{:,n}}) \left( \boldsymbol{X}^\top \boldsymbol{W}_Q^\top \boldsymbol{W}_K \otimes \boldsymbol{e}_j \right) \right\| \\
&\leq \frac{1}{\sqrt{d}} \|\boldsymbol{W}_V \boldsymbol{X}\| \max_i \left\| \frac{\partial \boldsymbol{A}_{:,i}}{\partial \boldsymbol{M}_{:,i}} \right\| \|\boldsymbol{X}^\top \boldsymbol{W}_Q^\top \boldsymbol{W}_K\| \\
&\leq \frac{5\varepsilon'}{\sqrt{d}} \|\boldsymbol{W}_V\|_2 \|\boldsymbol{W}_Q^\top \boldsymbol{W}_K\| \|\boldsymbol{X}\|^2,
\end{aligned}
$$

where in the last step we use the result in Lemma B.2, that is

$$\left\| \frac{\partial \mathbf{A}_{:,i}}{\partial \mathbf{M}_{:,i}} \right\| \le \sqrt{1 + \frac{4}{T-1}} \epsilon_i \le 5\epsilon'$$

As for (II), we follow the same computation

$$(\mathrm{II}) : \left\| \frac{1}{\sqrt{d}} \operatorname{diag}(\mathbf{W}_V \mathbf{X} \frac{\partial \mathbf{A}_{:,1}}{\partial \mathbf{M}_{:,1}}, \cdots, \mathbf{W}_V \mathbf{X} \frac{\partial \mathbf{A}_{:,n}}{\partial \mathbf{M}_{:,n}}) (\mathbf{e}_j \otimes \mathbf{X}^\top \mathbf{W}_K^\top \mathbf{W}_Q) \right\|$$

$$\le \frac{1}{\sqrt{d}} \|\mathbf{W}_V \mathbf{X}\| \max_i \left\| \frac{\partial \mathbf{A}_{:,i}}{\partial \mathbf{M}_{:,i}} \right\| \|\mathbf{X}^\top \mathbf{W}_K^\top \mathbf{W}_Q\|$$

$$\le \frac{5\varepsilon'}{\sqrt{d}} \|\mathbf{W}_V\| \|\mathbf{W}_K^\top \mathbf{W}_Q\| \|\mathbf{X}\|^2.$$

To bound (III), we have to treat the first token and others differently. Since most attention weights are concentrated on the first token, its corresponding (III) will be much larger. As mentioned in the main text, we mainly focus on the case where only one sink token is present. we assume that for any $k \in \{2, \cdots, T\}$ and $j \in \{2, \cdots, k\}$, $\mathbf{A}_{j,k} = \frac{\varepsilon_i}{T-1}$, that is, the 2-th to the $T$-th attention weights are uniform. This assumption is not restrictive by the conclusions in (Veličković et al., 2024). As a result, when $j = 1$, we have

$$\|\mathbf{A}_{1,:}\| = \sqrt{\sum_{i=1}^{T} (1 - \varepsilon_i)^2} \le \sqrt{T}(1 - \varepsilon),$$

When $j \ne 1$, similarly we have

$$\|\mathbf{A}_{j,:}\| = \sqrt{\sum_{i=1}^{T} \left( \frac{\epsilon_i}{T-1} \right)^2} \le \epsilon' \sqrt{\sum_{i=1}^{T} \left( \frac{1}{T-1} \right)^2} \le \frac{\sqrt{T}}{T-1} \varepsilon' \le \frac{\sqrt{2}}{\sqrt{T-1}} \varepsilon',$$

where we use the following fact

$$\frac{\sqrt{T}}{T-1} = \frac{\sqrt{T/T-1}}{\sqrt{T-1}} \le \frac{\sqrt{2}}{\sqrt{T-1}}, \quad \text{for any } T \ge 2.$$

Combining these two cases, we have

$$(\mathrm{III}) : \|\mathbf{A}_{j,:} \otimes \mathbf{W}_V\| = \|\mathbf{A}_{j,:}\| \|\mathbf{W}_V\| \le \begin{cases} \sqrt{T}(1 - \varepsilon)\|\mathbf{W}_V\| & \text{if } j = 1, \\ \frac{\sqrt{2}\varepsilon'}{\sqrt{T-1}} \|\mathbf{W}_V\| & \text{otherwise,} \end{cases}$$

Combining (I), (II) and (III), we have

$$\|\mathbf{J}_{:,j}\| \le \begin{cases} \|\mathbf{W}_V\| \cdot \left( \frac{10\varepsilon'}{\sqrt{d}} \|\mathbf{X}\|^2 \|\mathbf{W}_K^\top \mathbf{W}_Q\| + \sqrt{T}(1 - \epsilon) \right) & \text{if } j = 1, \\ \|\mathbf{W}_V\| \cdot \left( \frac{10\varepsilon'}{\sqrt{d}} \|\mathbf{X}\|^2 \|\mathbf{W}_K^\top \mathbf{W}_Q\| + \frac{\sqrt{2}\varepsilon'}{\sqrt{T-1}} \right) & \text{otherwise.} \end{cases}$$

We treat $d$ as constant and assume that $\|\mathbf{X}\|^2$, $\|\mathbf{W}_K^\top \mathbf{W}_Q\|$, $\|\mathbf{W}_V\|$ are bounded by a universal constant. Then we can find a sufficiently large $K_1, K_2, K_3 \in \mathbb{R}$ such that

$$\|\mathbf{J}_{:,j}\| \le \begin{cases} K_1 \sqrt{T}(1 - \varepsilon) + K_2 \varepsilon' & \text{if } j = 1, \\ K_1 \frac{\varepsilon'}{\sqrt{T-1}} + K_2 \varepsilon' & \text{otherwise,} \end{cases}$$

where

$$K_1 = \sqrt{2}\|\boldsymbol{W}_V\|,$$
$$K_2 = \|\boldsymbol{W}_V\| \cdot \frac{10}{\sqrt{d}}\|\boldsymbol{X}\|^2\|\boldsymbol{W}_K^\top \boldsymbol{W}_Q\|.$$

Since $\frac{\partial(\boldsymbol{S}+\boldsymbol{H})}{\partial \boldsymbol{h}_i}$ is the $i$-th column of the matrix in Eq.(4), we have

$$\left\|\frac{\partial(\boldsymbol{S}+\boldsymbol{H})}{\partial \boldsymbol{h}_i}\right\| = \left\|\boldsymbol{J}_{:,i}\frac{\partial \mathrm{LN}(\boldsymbol{h}_i)}{\partial \boldsymbol{h}_i} + \boldsymbol{I}_d\right\| \le \|\boldsymbol{J}_{:,i}\|\left\|\frac{\partial \mathrm{LN}(\boldsymbol{h}_i)}{\partial \boldsymbol{h}_i}\right\| + 1$$
$$\le \begin{cases} K_1\frac{\sqrt{T}(1-\varepsilon)}{\|\boldsymbol{h}_1\|} + K_2\frac{\varepsilon'}{\|\boldsymbol{h}_1\|} + 1 & \text{if } i = 1, \\ K_1\frac{(\sqrt{T-1})^{-1}\varepsilon'}{\|\boldsymbol{h}_i\|} + K_2\frac{\varepsilon'}{\|\boldsymbol{h}_i\|} + 1 & \text{otherwise,} \end{cases}$$

where in the second inequality, we use the fact in Lemma B.1 that

$$\left\|\frac{\mathrm{LN}(\boldsymbol{h}_i)}{\partial \boldsymbol{h}_i}\right\| \le 2\|\boldsymbol{g}\|_\infty\frac{\sqrt{d}}{\|\boldsymbol{h}_i\|},$$

where $\boldsymbol{g}$ is the scaling factor in LN module. As for the universal constant $K_1, K_2$, we bound them in the following way

$$K_1 = 2\sqrt{2d}\|\boldsymbol{g}\|_\infty\|\boldsymbol{W}_V\|,$$
$$K_2 = 20\|\boldsymbol{g}\|_\infty\|\boldsymbol{X}\|^2\|\boldsymbol{W}_K^\top \boldsymbol{W}_Q\|\|\boldsymbol{W}_V\|.$$

Then, we consider multi-head attention. Using the same notation in the main text, we have

$$\boldsymbol{H}^{\ell+1/2} = \mathrm{MHSA}(\mathrm{LN}(\boldsymbol{H}^\ell)) + \boldsymbol{H}^\ell,$$
$$\mathrm{MHSA}(\mathrm{LN}(\boldsymbol{H}^\ell)) = \sum_{i=1}^{H}\boldsymbol{W}_O^{\ell,i}\boldsymbol{W}_V^{\ell,i}\,\mathrm{Softmax}\left[(\boldsymbol{W}_K^{\ell,i}\mathrm{LN}(\boldsymbol{H}^\ell))^\top(\boldsymbol{W}_Q^{\ell,i}\mathrm{LN}(\boldsymbol{H}^\ell))\right],$$

where we replace the concatenation operation by a summation of matrices, and $\boldsymbol{W}_O^{\ell,i} \in \mathbb{R}^{d \times d_h}$, $\boldsymbol{W}_O^\ell = \left[\boldsymbol{W}_O^{\ell,1}, \cdots, \boldsymbol{W}_O^{\ell,H}\right]$. Then the following holds based on the our previous analysis

$$\left\|\frac{\partial \boldsymbol{H}^{\ell+1/2}}{\partial \boldsymbol{h}_i^\ell}\right\| \le \begin{cases} K_1\frac{\sqrt{T}(1-\varepsilon)}{\|\boldsymbol{h}_1^\ell\|} + K_2\frac{\varepsilon'}{\|\boldsymbol{h}_1^\ell\|} + 1 & \text{if } i = 1, \\ K_1\frac{(\sqrt{T-1})^{-1}\varepsilon'}{\|\boldsymbol{h}_i^\ell\|} + K_2\frac{\varepsilon'}{\|\boldsymbol{h}_i^\ell\|} + 1 & \text{otherwise,} \end{cases}$$

where

$$K_1 = O(\sqrt{d}\sum_{i=1}^{H}\|\boldsymbol{g}\|_\infty\|\boldsymbol{W}_V^{\ell,i}\|\|\boldsymbol{W}_O^{\ell,i}\|),$$
$$K_2 = O(d\sum_{i=1}^{H}\|\boldsymbol{W}_V^{\ell,i}\|\|\boldsymbol{g}\|_\infty\|(\boldsymbol{W}_K^{\ell,i})^\top \boldsymbol{W}_Q^{\ell,i}\|),$$

which completes the proof.

$$\square$$

## B.4. Proof of Lemma B.1

In the following theorem, we compute $\left\| \frac{\mathrm{LN}(x)}{\partial x} \right\|$, which scales inversely with $\|x\|$.

**Lemma B.1.** *Let* $\mathrm{LN}(\cdot)$ *denote RMSNorm, then the following holds for any* $x \in \mathbb{R}^d$.

$$\left\| \frac{\partial \mathrm{LN}(x)}{\partial x} \right\| \leq \max_i \{|g_i|\} \frac{\sqrt{d}}{\|x\|}.$$

*Proof.* For generality and due to the similarity between layer normalization and RMSNorm, we begin with layer normalization first. The definition of layer normalization is

$$\mathrm{LN}(x)_i = \frac{x_i - \mu}{\sigma} \cdot g_i + b_i,$$

$$\mu = \frac{1}{d} \sum_{i=1}^d x_i,$$

$$\sigma = \sqrt{\frac{1}{d} \sum_{i=1}^d (x_i - \mu)^2}.$$

Let $g = (g_1, \cdots, g_d)$, $y = x(I - \frac{1}{d}\mathbf{1}\mathbf{1}^\top)$, then $\mathrm{LN}(x)_i = \frac{y_i}{\sqrt{\frac{1}{d}\sum_{i=1}^d y_i^2}} \cdot g_i + b_i$. We can compute

$$
\begin{aligned}
\frac{\partial \mathrm{LN}(x)_i}{\partial y_j} &= \frac{\partial}{\partial y_j} \left( \frac{y_i}{\sqrt{\frac{1}{d}\sum_{i=1}^d y_i^2}} \cdot g_i + b_i \right) \\
&= g_i \cdot \frac{\delta_{ij}\sqrt{\frac{1}{d}\sum_{k=1}^d y_k^2} - y_i \frac{\frac{1}{d}y_j}{\sqrt{\frac{1}{d}\sum_{k=1}^d y_k^2}}}{\frac{1}{d}\sum_{k=1}^d y_k^2} \\
&= g_i\sqrt{d}\frac{\delta_{ij}\|y\|^2 - y_i y_j}{\|y\|^{3/2}} \\
&= g_i\sqrt{d}\left( \delta_{ij} - \frac{y_i y_j}{\|y\|^2} \right).
\end{aligned}
$$

Then, rewrite it in the matrix form, we have

$$\frac{\partial \mathrm{LN}(x)}{\partial y} = \mathrm{diag}(g) \cdot \frac{\sqrt{d}}{\|y\|} \left( I - \frac{yy^\top}{\|y\|^2} \right).$$

Then

$$
\begin{aligned}
\frac{\partial \mathrm{LN}(x)}{\partial x} &= \frac{\partial \mathrm{LN}(x)}{\partial y} \cdot \frac{\partial y}{\partial x} \\
&= \mathrm{diag}(g) \cdot \frac{\sqrt{d}}{\|y\|} \left( I - \frac{yy^\top}{\|y\|^2} \right) \left( I - \frac{1}{d}\mathbf{1}\mathbf{1}^\top \right).
\end{aligned}
$$

As for RMSNorm, it discards the centering step, that is, $\mathrm{LN}(x)_i = \frac{x_i}{\sqrt{\frac{1}{d}\sum_{k=1}^d x_k^2}} \cdot g_i$. Following the same computation above, we have

$$\frac{\partial \mathrm{LN}(x)}{\partial x} = \mathrm{diag}(g) \cdot \frac{\sqrt{d}}{\|x\|} \left( I - \frac{xx^\top}{\|x\|^2} \right).$$

For any real symmetric matrix $A \in \mathbb{R}^{n \times n}$, it can be written as $A = Q\Lambda Q^\top$, where $Q \in \mathbb{R}^{n \times n}$ is an orthogonal matrix and $\Lambda = \mathrm{diag}(\lambda_1, \cdots, \lambda_n)$ ($\lambda_1, \cdots, \lambda_n$ are the eigenvalues of $A$). Therefore, $A^\top A = A^2 = \left( Q\Lambda Q^\top \right)\left( Q\Lambda Q^\top \right) =$

$Q\Lambda^2 Q^\top$, which implies that the eigenvalues of $A^2$ are $\lambda_1^2, \cdots, \lambda_n^2$. Since each singular value $\sigma_i$ of $A$ is one of the eigenvalues of $A^\top A$, then we have $\sigma_i = \sqrt{\lambda_i^2} = |\lambda_i|$. According to Lemma 4 in (Xiong et al., 2020), for any vector $\alpha \in \mathbb{R}^d$ such that $\|\alpha\|_2 = 1$, then the eigenvalues of $I - \alpha\alpha^\top$ are either 1 or 0. For layer normalization, we have

$$\left\|\frac{\partial \mathrm{LN}(x)}{\partial x}\right\| \leq \frac{\sqrt{d}}{\|y\|}\|\operatorname{diag}(g)\| \cdot \left\|I - \frac{yy^\top}{\|y\|^2}\right\| \cdot \left\|I - \frac{1}{d}\mathbf{1}\mathbf{1}^\top\right\| = \frac{\sqrt{d}}{\|y\|}\|\operatorname{diag}(g)\| \leq \max_i\{|g_i|\}\frac{\sqrt{d}}{\|y\|}.$$

If we further assume that $x$ is highly sparse, then we have

$$\left\|\frac{\partial \mathrm{LN}(x)}{\partial x}\right\| \leq 2\max_i\{|g_i|\}\frac{\sqrt{d}}{\|x\|}.$$

Similarly, for RMSNorm we also have

$$\left\|\frac{\partial \mathrm{LN}(x)}{\partial x}\right\| \leq \frac{\sqrt{d}}{\|x\|}\|\operatorname{diag}(g)\|\left\|I - \frac{xx^\top}{\|x\|^2}\right\| \leq \max_i\{|g_i|\}\frac{\sqrt{d}}{\|x\|},$$

which completes the proof.

$\square$

### B.5. Proof of Lemma B.2

**Lemma B.2.** *For any $x \in \mathbb{R}^d$ with $d > 1$, let $s = \mathrm{Softmax}(x) := [s_1, \cdots, s_d]$ with $s_1 = 1 - \varepsilon$ for some $\varepsilon \in (0, 1)$, then we have*

$$\|J(s)\|_F = \left\|\frac{\partial \operatorname{softmax}(x)}{\partial x}\right\|_F \leq \sqrt{1 + \frac{4}{d-1}} \cdot \varepsilon.$$

*Proof.* Let $J(s) = \frac{\partial s}{\partial x}$. It is clear that $J^\top = J$. Then we have

$$\|J(s)\|_F^2 = \mathrm{tr}(J^\top J) = \mathrm{tr}(J^2).$$

Through direct computation, we have the following results

$$J^2 = \left(\operatorname{diag}(s) - ss^\top\right)^2 = \operatorname{diag}(s)^2 - 2\operatorname{diag}(s)ss^\top + (ss^\top)^2$$

$$\mathrm{tr}(\operatorname{diag}(s)^2) = \sum_{i=1}^d s_i^2$$

$$\mathrm{tr}(\operatorname{diag}(s)ss^\top) = \mathrm{tr}(s^\top \operatorname{diag}(s)s) = \sum_{i=1}^d s_i^3$$

$$\mathrm{tr}((ss^\top)^2) = \mathrm{tr}(ss^\top ss^\top) = \mathrm{tr}(s^\top ss^\top s) = \mathrm{tr}((s^\top s)^2) = (s^\top s)^2 = \left(\sum_{i=1}^d s_i^2\right)^2.$$

Combining these results we have

$$\|J\|_F^2 = \sum_{i=1}^d s_i^2 - 2\sum_{i=1}^d s_i^3 + \left(\sum_{i=1}^d s_i^2\right)^2.$$

Our goal is to solve the following maximization problem

$$\max_s \sum_{i=1}^d s_i^2 - 2\sum_{i=1}^d s_i^3 + \left(\sum_{i=1}^d s_i^2\right)^2 \quad \text{s.t } s_1 = 1 - \varepsilon, \ \sum_{i=2}^d s_i = \varepsilon, s_i > 0.$$

Define the Lagrangian as

$$\mathcal{L}(s_2, \cdots, s_d, \lambda) = (1-\varepsilon)^2 + \sum_{i=2}^{d} s_i^2 - 2(1-\varepsilon)^3 - 2\sum_{i=2}^{d} s_i^3 + \left((1-\varepsilon)^2 + \sum_{i=2}^{d} s_i^2\right)^2 - \lambda\left(\sum_{i=2}^{d} s_i - \varepsilon\right).$$

Subsequently, we solve the following equations

$$\frac{\partial \mathcal{L}(s_2, \cdots, s_d, \lambda)}{\partial s_i} = 2s_i - 6s_i^2 + 2\left((1-\varepsilon)^2 + \sum_{i=2}^{d} s_i^2\right)2s_i - \lambda = 0$$

$$\implies 2s_i - 6s_i^2 + 4s_i\left((1-\varepsilon)^2 + \sum_{i=2}^{d} s_i^2\right) = \lambda \quad \text{for } i = 2, \cdots, d.$$

Then, we know that $s_i = s_j$ for any $i \neq j \in \{2, \cdots, d\}$. Based on the constraint $\sum_{i=2}^{d} s_i = \varepsilon$, we have

$$s_2 = \cdots = s_d = \frac{\varepsilon}{d-1}.$$

Put this solution into the computation of $\|\boldsymbol{J}\|_F^2$, we have

$$\max_{\boldsymbol{s}} \|\boldsymbol{J}\|_F^2 = (1-\varepsilon)^2 + \frac{\varepsilon^2}{d-1} - 2\left((1-\varepsilon)^3 + \frac{\varepsilon^3}{(d-1)^2}\right) + \left((1-\varepsilon)^2 + \frac{\varepsilon^2}{d-1}\right)^2.$$

In the following, we give an upper bound of $\max_{\boldsymbol{s}} \|\boldsymbol{J}\|_F^2$ in terms of $\varepsilon$.

$$(1-\varepsilon)^2 + \frac{\varepsilon^2}{d-1} - 2\left((1-\varepsilon)^3 + \frac{\varepsilon^3}{(d-1)^2}\right) + \left((1-\varepsilon)^2 + \frac{\varepsilon^2}{d-1}\right)^2$$

$$= \left[(1-\varepsilon)^2 - 2(1-\varepsilon)^3 + (1-\varepsilon)^4\right] + \frac{\varepsilon^2}{d-1} - \frac{2\varepsilon^3}{d-1} + \frac{2\varepsilon^2(1-\varepsilon)^2}{d-1} + \frac{\varepsilon^4}{d-1}$$

$$\leq \left[(1-\varepsilon)^2 - 2(1-\varepsilon)^3 + (1-\varepsilon)^4\right] + \frac{\varepsilon^2}{d-1} + \frac{2\varepsilon^2}{d-1} + \frac{\varepsilon^2}{d-1}$$

$$= \left[(1-\varepsilon)^2 - 2(1-\varepsilon)^3 + (1-\varepsilon)^4\right] + \frac{4\varepsilon^2}{d-1}.$$

To bound $(1-\varepsilon)^2 - 2(1-\varepsilon)^3 + (1-\varepsilon)^4$, we have

$$(1-\varepsilon)^2 - 2(1-\varepsilon)^3 + (1-\varepsilon)^4 = 1 - 2\varepsilon + \varepsilon^2 - 2(1 - 3\varepsilon + 3\varepsilon^2 - \varepsilon^3) + (1 - 4\varepsilon + 6\varepsilon^2 - 4\varepsilon^3 + \varepsilon^4)$$

$$= \varepsilon^4 - 2\varepsilon^3 + \varepsilon^2 \leq \varepsilon^2,$$

where we use the fact $\varepsilon^4 - 2\varepsilon^3 \leq 0$ for any $\varepsilon \in (0, 1)$. Then we have

$$\max_{\boldsymbol{s}} \|\boldsymbol{J}\|_F^2 \leq \left(1 + \frac{4}{d-1}\right)\varepsilon^2.$$

The proof is completed by computing the square root on both sides. $\square$

## C. Sparse and Uniform Massive Activations

Due to the close relationship between massive activations and RMSNorm, we show that a large hidden-state norm $|\boldsymbol{h}_1^\ell|$ is necessary to counteract the imbalance in token mixing induced by attention sinks. However, the mechanisms for producing a large $|\boldsymbol{h}_1^\ell|$ can vary across models. As observed in (Sun et al., 2024), massive activations are highly sparse in the hidden states of sink tokens, typically appearing in fewer than five dimensions. Some researchers hypothesize that these massive activations are preserved after RMSNorm, while the remaining dimensions are effectively suppressed. Consequently, the keys and values of sink tokens exhibit a geometry that differs substantially from that of non-sink tokens, thereby facilitating the formation of attention sinks.

However, this explanation leaves an important question unresolved. Since RMSNorm is approximately scaling-invariant, i.e., scaling the hidden states by a small factor does not significantly change the outputs, it is unclear why models still prefer to learn extremely large activations in sink tokens and distinguish them so sharply from non-sink tokens. Our result in Theorem 4.1 clarifies the fundamental reason why massive activations emerge in sink tokens. Although we do not rule out the possibility that the sparsity pattern of these activations serves additional functional roles (for example, we find that in Qwen3-8B, the directions of the hidden states induced by the sparsity of massive activations are important, whereas they appear to be less important in the LLaMA family), our analysis answers why the activations themselves have to be extremely large.

As presented in Section 6, we zero out the top-$k$ entries in the output of each RMSNorm and observe that attention sinks are largely preserved, with only a slight degradation as $k$ increases. This finding suggests that the sparsity of massive activations may not be essential for the formation of attention sinks. Since our experiments are primarily conducted on the LLaMA family, this conclusion may not generalize to other model families. Indeed, although attention sinks have been observed across a wide range of models, the underlying mechanisms responsible for their formation may differ substantially.

We further observe that, in (Park et al., 2025), the trained LLMs ("dmis-lab/OSP-1.4B-1T-Muon-SSNorm-EmbProj") also assign significantly larger hidden-state norms to sink tokens, but the corresponding massive activations are distributed much more uniformly across the embedding dimensions rather than being concentrated in only a few positions. The visualization is shown in Figure 13. As illustrated in Figure 13a, sink tokens still exhibit dominant hidden-state norms across layers. However, Figure 13b shows that these large activations are distributed far more uniformly across dimensions compared to the highly sparse patterns observed in Figure 8.

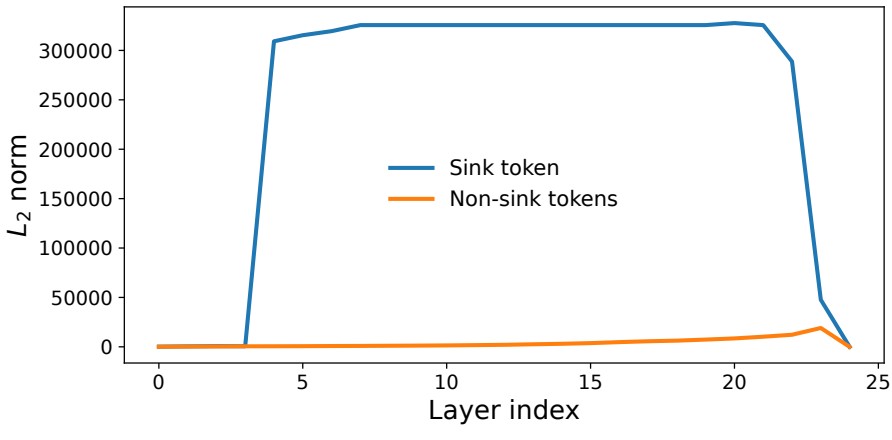

*(a)* The norm of hidden states of sink and non-sin tokens in (Park et al., 2025).

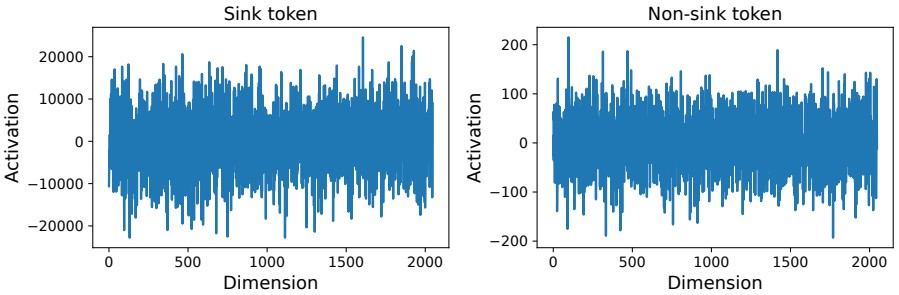

*(b)* The visualization of hidden states.

*Figure 13.* Massive activations in the models trained by (Park et al., 2025).

## D. Functional Differences Between FFN and Self-Attention

Note that Theorem 4.1 focuses exclusively on self-attention layers, where attention sinks appear. We have shown that $\left\| \frac{\partial \boldsymbol{H}^{\ell+1/2}}{\partial \boldsymbol{H}^\ell} \right\|$ can be well controlled by both attention sinks and massive activations. However, how each FFN can influence

token mixing has not been studied. We consider the FFN in the $\ell$-th Transformer block and the self-attention in the $(\ell+1)$-th Transformer block, then we have

$$\boldsymbol{H}^{\ell+1/2} = \mathrm{MHSA}(\mathrm{LN}(\boldsymbol{H}^{\ell})) + \boldsymbol{H}^{\ell},$$
$$\boldsymbol{H}^{\ell} = \mathrm{FFN}(\mathrm{LN}(\boldsymbol{H}^{\ell-1/2})) + \boldsymbol{H}^{\ell-1/2}.$$

We aim to bound $\left\|\frac{\partial \boldsymbol{H}^{\ell+1/2}}{\partial \boldsymbol{h}_1^{\ell-1/2}}\right\|$, which satisfies $\left\|\frac{\partial \boldsymbol{H}^{\ell+1/2}}{\partial \boldsymbol{H}^{\ell}}\right\| \left\|\frac{\partial \boldsymbol{H}^{\ell}}{\partial \boldsymbol{h}_1^{\ell-1/2}}\right\|$. Since we already bound $\left\|\frac{\partial \boldsymbol{H}^{\ell+1/2}}{\partial \boldsymbol{H}^{\ell}}\right\|$ in the main text, it suffices to bound $\left\|\frac{\partial \boldsymbol{H}^{\ell}}{\partial \boldsymbol{h}_1^{\ell-1/2}}\right\|$. As each FFN possess tokens independently, we only need to bound $\left\|\frac{\partial \boldsymbol{h}_1^{\ell}}{\partial \boldsymbol{h}_1^{\ell-1/2}}\right\|$. By the relationship between $\boldsymbol{h}_1^{\ell}$ and $\boldsymbol{h}_1^{\ell-1/2}$, we have

$$\boldsymbol{h}_1^{\ell} = \mathrm{FFN}\left(\mathrm{LN}\left(\boldsymbol{h}_1^{\ell-1/2}\right)\right) + \boldsymbol{h}_1^{\ell-1/2}.$$

In this section, we mainly consider the case where the activation functions in each FFN module is SwiGLU function, which has the following form:

$$\mathrm{FFN}(\boldsymbol{x}) = \boldsymbol{W}_3 \left(\phi\left(\boldsymbol{W}_1 \boldsymbol{x}\right) \odot \left(\boldsymbol{W}_2 \boldsymbol{x}\right)\right),$$

where $\phi(x) = \frac{x}{1+e^{-x}}$ is the element-wise activation function. Through direct computation, we have

$$\frac{\partial \mathrm{FFN}(\boldsymbol{x})}{\partial \boldsymbol{x}} = \frac{\partial \mathrm{FFN}(\boldsymbol{x})}{\boldsymbol{W}_1 \boldsymbol{x}} \frac{\partial \boldsymbol{W}_1 \boldsymbol{x}}{\partial \boldsymbol{x}} + \frac{\partial \mathrm{FFN}(\boldsymbol{x})}{\boldsymbol{W}_2 \boldsymbol{x}} \frac{\partial \boldsymbol{W}_2 \boldsymbol{x}}{\partial \boldsymbol{x}}$$
$$= \boldsymbol{W}_3 \left(\mathrm{diag}(\boldsymbol{W}_2 \boldsymbol{x}) \odot \mathrm{diag}\left(\frac{\partial \phi(\boldsymbol{x})}{\partial \boldsymbol{x}}\right) + \mathrm{diag}\left(\phi(\boldsymbol{W}_1 \boldsymbol{x})\right)\right).$$

To bound its spectral norm, we have

$$\left\|\frac{\partial \mathrm{FFN}(\boldsymbol{x})}{\partial \boldsymbol{x}}\right\| \le \|\boldsymbol{W}_3\| \left(\left\|\mathrm{diag}(\boldsymbol{W}_2 \boldsymbol{x}) \odot \mathrm{diag}\left(\frac{\partial \phi(\boldsymbol{x})}{\partial \boldsymbol{x}}\right)\right\| + \|\mathrm{diag}(\phi(\boldsymbol{W}_1 \boldsymbol{x}))\|\right).$$

To bound first term, we have

$$\left\|\mathrm{diag}(\boldsymbol{W}_2 \boldsymbol{x}) \odot \mathrm{diag}\left(\frac{\partial \phi(\boldsymbol{x})}{\partial \boldsymbol{x}}\right)\right\| \le \|\mathrm{diag}(\boldsymbol{W}_2 \boldsymbol{x})\| \left\|\mathrm{diag}\left(\frac{\partial \phi(\boldsymbol{x})}{\partial \boldsymbol{x}}\right)\right\| \le 2\|\boldsymbol{W}_2\|\|\boldsymbol{x}\|,$$

where we use the fact that $\frac{\partial \phi(x)}{\partial x} \in (-2.8, 1.1)$. As for the second term, we bound it in the following way

$$\|\mathrm{diag}(\phi(\boldsymbol{W}_1 \boldsymbol{x}))\| \le \|\boldsymbol{W}_1\|\|\boldsymbol{x}\|.$$

Then we have

$$\left\|\frac{\partial \mathrm{FFN}(\mathrm{LN}(\boldsymbol{x}))}{\partial \mathrm{LN}(\boldsymbol{x})} \frac{\partial \mathrm{LN}(\boldsymbol{x})}{\partial \boldsymbol{x}}\right\| \le (2\|\boldsymbol{W}_2\|\|\mathrm{LN}(\boldsymbol{x})\| + \|\boldsymbol{W}_1\|\|\mathrm{LN}(\boldsymbol{x})\|) \cdot \max_i\{|g_i|\} \frac{\sqrt{d}}{\|\boldsymbol{x}\|}$$
$$\le \frac{K_3}{\|\boldsymbol{x}\|},$$

where we assume $\|\mathrm{LN}(\boldsymbol{x})\|$ is bounded by a universal constant and absorb all constant terms into $K_3$. By taking the skip connection into consideration, we have

$$\left\|\frac{\partial \boldsymbol{h}_1^{\ell}}{\partial \boldsymbol{h}_1^{\ell-1/2}}\right\| \le \frac{K_3}{\|\boldsymbol{h}_1^{\ell-1/2}\|} + 1.$$

As a result, by combining Theorem 4.1, we have the following bound

$$\left\|\frac{\partial \boldsymbol{H}^{\ell+1/2}}{\partial \boldsymbol{h}_1^{\ell-1/2}}\right\| \leq \left(K_1 \frac{\sqrt{T-1}\varepsilon}{\|\boldsymbol{h}_1^{\ell}\|} + K_2 \frac{\epsilon'}{\|\boldsymbol{h}_1^{\ell}\|} + 1\right)\left(\frac{K_3}{\|\boldsymbol{h}_1^{\ell-1/2}\|} + 1\right).$$

The above bound shows that FFNs may amplitude the value of massive activations.

One more insight is that the above bound shows that only $\left\|\frac{\partial \boldsymbol{h}_1^{\ell}}{\partial \boldsymbol{h}_1^{\ell-1/2}}\right\|$ is relatively small, while other tokens remain stable. To investigate role of FFN modules empirically, we examine the difference between hidden states of the same token under different contextual inputs. As illustrated in Figure 14, we compute the $L_2$-norm of the difference between representations of the same token across two contexts (we fix the sequence length and make the last token be the same), focusing on the layer input, the output of each self-attention block, and the output of each FFN block. Notably, when the two contexts are either highly similar or markedly different, the curves corresponding to the layer input (yellow), and the self-attention output (blue) largely overlap. This observation indicates that the quantity $\left\|\frac{\partial \boldsymbol{H}^{\ell+1/2}}{\partial \boldsymbol{H}^{\ell}}\right\|$ remains relatively small and FFN layers contribute most to produce the context-dependent representations.

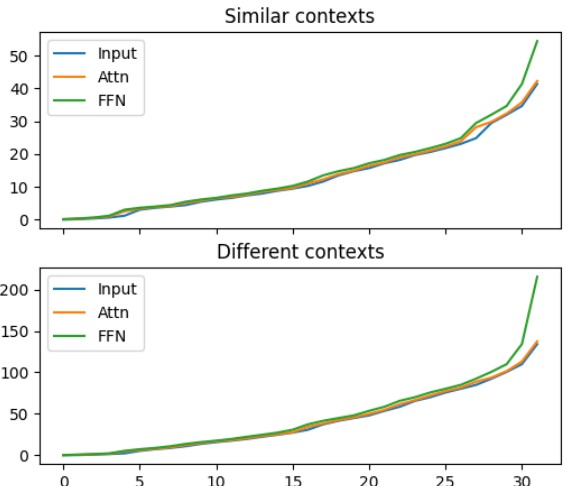

*Figure 14.* **Difference in hidden states of the same token across contexts and layers.** We consider three input sentences whose final tokens are identical. The first two sentences share similar contextual prefixes, whereas the third sentence is embedded in a distinctly different context. Let $\boldsymbol{H}^{i,\ell}$, for $i = 1, 2, 3$, denote the hidden state matrix at layer $\ell$ corresponding to the $i$-th input. At each layer, we compute the $L_2$-norm of the difference between the representations of the final token across different inputs at three stages of the layer: the layer input, the output of the self-attention block, and the output of the FFN block. Specifically, we evaluate $\|\boldsymbol{H}_{:,-1}^{i,\ell-1} - \boldsymbol{H}_{:,-1}^{j,\ell-1}\|$, $\|\boldsymbol{H}_{:,-1}^{i,\ell-1/2} - \boldsymbol{H}_{:,-1}^{j,\ell-1/2}\|$ and $\|\boldsymbol{H}_{:,-1}^{i,\ell} - \boldsymbol{H}_{:,-1}^{j,\ell}\|$, which correspond to the blue, yellow, and green curves in the figure, respectively.

**Connections to the ineffectiveness of deeper layers.** Some works (Gromov et al., 2025) observed that removing a large fraction of layers in LLMs only leads to only minimal degradation on downstream tasks. Razzhigaev et al. (2024) demonstrated that embedding transformations between sequential layers in Transformer decoders exhibit almost linear properties. Queipo-de Llano et al. (2025) proposed a Mix-Compress-Refine theory of information flow, explaining how LLMs organize computation in depth and why different tasks tend to use different effective depth. Li et al. (2025); Sun et al. (2025) attributed this phenomenon to the widely used pre-LN in LLMs and propose several mitigation strategies. To the best of our knowledge, few works have explicitly explored the connection between the ineffectiveness of deeper layers and the phenomena of attention sink or massive activations.

We point out that these phenomena shares a closely related mathematical formulation. Specifically, Theorem 5.1 establishes an explicit upper bound on $\left\|\frac{\partial \boldsymbol{H}^{\ell+1/2}}{\partial \boldsymbol{H}^{\ell}}\right\|$ in terms of attention sink and massive activations, showing that under these conditions, the self-attention layer behaves increasingly like a pure skip connection. This characterization aligns with the observation that self-attention layers with attention sink contribute little additional transformation to the hidden representations (Bondarenko et al., 2023). A related theoretical result appears in Theorem 3.3 of (Sun et al., 2025), where it is shown that if the variance

of each hidden state grows exponentially with the layer index, then $\left\|\frac{\partial \boldsymbol{H}^L}{\partial \boldsymbol{H}^0}\right\|$ is bounded by a constant at initialization, implying that the output becomes insensitive to the input. Intuitively, if the hidden states become excessively large, the pre-LN modules may render them insensitive to updates from subsequent layers.

As shown in Figure 8, the hidden states of non-sink tokens remain small, whereas those of sink tokens become significantly larger. This suggests that the presence of attention sinks causes non-sink tokens to undergo only mild updates during each self-attention operation, while FFNs may remain largely unaffected. In contrast, for sink tokens, the existence of massive activations makes them insensitive to updates from both self-attention and FFNs. However, it remains unclear whether attention sinks and massive activations work together to primarily contribute to the ineffectiveness of deeper layers.

## E. Effect of Massive Activations on Training Stability

In the following, we utilize the same notation in Appendix B. As shown in the proof of Theorem 4.1, we have

$$\frac{\partial \boldsymbol{S}}{\partial \boldsymbol{X}} = \frac{1}{\sqrt{d}}(\boldsymbol{I}_n \otimes \boldsymbol{W}_V \boldsymbol{X})\frac{\partial \boldsymbol{A}}{\partial \boldsymbol{M}}\left((\boldsymbol{X}^\top \boldsymbol{W}_Q^\top \boldsymbol{W}_K \otimes \boldsymbol{I}_n)\boldsymbol{K}_{dn} + (\boldsymbol{I}_n \otimes \boldsymbol{X}^\top \boldsymbol{W}_K^\top \boldsymbol{W}_Q)\right) + \boldsymbol{A}^\top \otimes \boldsymbol{W}_V.$$

Motivated by (Yudin et al., 2025), we have the following bound

$$\left\|\frac{\partial \boldsymbol{S}}{\partial \boldsymbol{X}}\right\| \leq \|\boldsymbol{W}_V\| \left(\|\boldsymbol{A}\| + 2\|\boldsymbol{X}\|^2\|\boldsymbol{W}_K^\top \boldsymbol{W}_Q\| \max_i \left\|\frac{\partial \boldsymbol{A}_{:,i}}{\partial \boldsymbol{M}_{:,i}}\right\|\right).$$

So far, we observe a clear trade-off between $\|\boldsymbol{A}\|$ and $\max_i \left\|\frac{\partial \boldsymbol{A}_{:,i}}{\partial \boldsymbol{M}_{:,i}}\right\|$. In particular, if each column of $\boldsymbol{A}$ is a one-hot vector, $\|\boldsymbol{A}\| = \sqrt{n}$ while $\max_i \left\|\frac{\partial \boldsymbol{A}_{:,i}}{\partial \boldsymbol{M}_{:,i}}\right\| = 0$, which are their maximum and minimum values respectively. This property motivates the following discussion on entropy collapse.

**Connections to entropy collapse and training stability.** For any attention probability matrix $\boldsymbol{A}$ defined in eq.(1), the attention entropy of the $i$-th column is defined as

$$H(\boldsymbol{A}_{:,i}) = -\sum_{j=1}^{T} \boldsymbol{A}_{j,i} \log \boldsymbol{A}_{j,i},$$

and the average attention entropy over all columns is given by

$$H(\boldsymbol{A}) = \frac{1}{T}\sum_{i=1}^{T} H(\boldsymbol{A}_{:,i}).$$

It is straightforward to verify that $H(\boldsymbol{A}_{:,i})$ attains its maximum when the attention weights are uniform, i.e., $\boldsymbol{A}_{j,i} = \frac{1}{T}$ for any $j = 1, \cdots, T$, and its minimum when $\boldsymbol{A}_{:,i}$ is a one-hot vector. Entropy collapse refers to the phenomenom in which low attention entropy emerges during training and is accompanied by high training instability (Zhai et al., 2023; Hong & Lee, 2025). In the presence of attention sinks, all tokens attend predominantly to the first token, resulting in low entropy in the attention probability matrix. However, despite this entropy collapse, prior empirical studies have not reported this training instability in such settings (Gu et al., 2021). Our result in Theorem 4.1 provides a possible explanation for apparent discenpancy: the coexistence of massive activations and attention sink effectively constrains the scale of gradients and prevents gradient spikes, thereby promoting training stability even under low attention entropy. Nevertheless, we emphasize that Theorem 4.1 does not characterize the training dynamics leading to the formation of attention sink and massive activations. Moreover, the extent to which massive activations influence the pre-training process remains unclear (Qiu et al., 2025).

## F. KV Biases, Gating Mechanism, Layer Normalization and Activation Functions

In this section, we provide a detailed discussion about how KV Biases, gating mechanism, layer normalization can affect the emergence of attention sinks and massive activations.

### F.1. KV Biases

Let's begin with KV biases. We conduct an in-depth analysis of several types of KV biases proposed in (Sun et al., 2024; Gu et al., 2021). Our primary focus is on the presence or absence of massive activations under these biasing schemes. We largely follow the notation introduced in (Gu et al., 2021), which may differ from that used in Section 3. The detailed analyses are presented below.

**(1)** Sun et al. (2024); Gu et al. (2024): $\text{Softmax}\left(\frac{1}{\sqrt{d_h}}\boldsymbol{Q}^{\ell,h}\left[\boldsymbol{k}^{*\ell,h^\top} \quad \boldsymbol{K}^{\ell,h^\top}\right] + \boldsymbol{M}\right)\begin{bmatrix}\boldsymbol{v}^{*\ell,h}\\\boldsymbol{V}^{\ell,h}\end{bmatrix}$ . This type of KV bias is analyzed in Section 4. As discussed there, since no gradients are propagated to the hidden states of the sink token, massive activations are not expected to emerge, consistent with the empirical findings in (Sun et al., 2024; Gu et al., 2024). A crucial prerequisite, however, is that the majority of attention scores are assigned to the corresponding biased key .

**(2)** Gu et al. (2024): $\text{Softmax}\left(\frac{1}{\sqrt{d_h}}\begin{bmatrix}\boldsymbol{q}^{*\ell,h}\\\boldsymbol{Q}^{\ell,h}\end{bmatrix}\left[\boldsymbol{k}^{*\ell,h^\top} \quad \boldsymbol{K}^{\ell,h^\top}\right] + \boldsymbol{M}\right)\begin{bmatrix}\boldsymbol{v}^{*\ell,h}\\\boldsymbol{V}^{\ell,h}\end{bmatrix}$ . Similar to the analysis in (1), $\boldsymbol{k}^{*\ell,h^\top}$ attracts the majority of attention scores, and consequently no massive activations are expected to emerge. See the empirical validation in Table 4 and Figure 7 of (Gu et al., 2024).

**(3)** Gu et al. (2024): $\text{Softmax}\left(\frac{1}{\sqrt{d_h}}\boldsymbol{Q}^{\ell,h}\left[\boldsymbol{k}^{*\ell,h^\top} \quad \boldsymbol{K}^{\ell,h^\top}\right] + \boldsymbol{M}\right)\begin{bmatrix}\boldsymbol{0}\\\boldsymbol{V}^{\ell,h}\end{bmatrix}$ . The biased key $\boldsymbol{k}^{*\ell,h^\top}$ itself is sufficient to attract the majority of attention scores and induce an attention sinks, consistent with the conclusions drawn in Section 6. Consequently, no massive activations are expected to emerge. This observation is empirically validated in Table 4 and Figure 7 of (Gu et al., 2024).

**(4)** Sun et al. (2024); Gu et al. (2024): $\text{Softmax}\left(\frac{1}{\sqrt{d_h}}\boldsymbol{Q}^{\ell,h}\boldsymbol{K}^{\ell,h^\top} + \boldsymbol{M}\right)\boldsymbol{V}^{\ell,h} + \boldsymbol{v}^{*\ell,h}$ . In the absence of KV biases and with only a value bias $\boldsymbol{v}^{*\ell,h}$, the model tends to select the first token as the sink token, leading to the emergence of massive activations on that token. This behavior is empirically validated in Table 4 and Figure 7 of (Gu et al., 2024), as well as Figure 40 of (Sun et al., 2024).

**(5)** Sun et al. (2024): $\text{Softmax}\left(\frac{1}{\sqrt{d_h}}\begin{bmatrix}\boldsymbol{q}' & \boldsymbol{Q}^{\ell,h}\end{bmatrix}\begin{bmatrix}\boldsymbol{k}' & \boldsymbol{K}^{\ell,h}\end{bmatrix}^\top + \boldsymbol{M}\right)\boldsymbol{V}^{\ell,h}$ . Inserting one extra feature dimension to both query and key matrix such that $\begin{bmatrix}\boldsymbol{q}' & \boldsymbol{Q}^{\ell,h}\end{bmatrix} \in \mathbb{R}^{T\times(d+1)}$ and $\begin{bmatrix}\boldsymbol{k}' & \boldsymbol{K}^{\ell,h}\end{bmatrix} \in \mathbb{R}^{T\times(d+1)}$. Under this setting, the model tends to select the first token as the sink token, leading to the emergence of massive activations. See the empirical validation in Figure 40 of (Sun et al., 2024).

**(6)** Sun et al. (2024): $\text{Softmax}\left(\frac{1}{\sqrt{d_h}}\boldsymbol{Q}^{\ell,h}\left[\boldsymbol{0}^\top \quad \boldsymbol{K}^{\ell,h^\top}\right] + \boldsymbol{M}\right)\begin{bmatrix}\boldsymbol{0}\\\boldsymbol{V}^{\ell,h}\end{bmatrix}$ . Replacing the KV biases with a constant vector is generally insufficient for the key bias to attract the majority of attention scores. As a result, the model may select the first token as the sink token, leading to the emergence of massive activations. See Figure 40 of (Sun et al., 2024) for more details.

**(7)** $\text{Softmax}\left(\frac{1}{\sqrt{d_h}}\boldsymbol{Q}^{\ell,h}\left[\boldsymbol{0}^\top \quad \boldsymbol{K}^{\ell,h^\top}\right] + \boldsymbol{M}\right)\begin{bmatrix}\boldsymbol{v}^{*\ell,h}\\\boldsymbol{V}^{\ell,h}\end{bmatrix}$ . To the best of our knowledge, this setting has not been examined in the existing literature. Motivated by the observation in Figure 7a—where zeroing out the hidden state restores the attention sinks—we conjecture that massive activations may not arise under this setting. Moreover, the functional role of $\boldsymbol{v}^{*\ell,h}$ could be more significant in the early layers, rather than in the later layers where attention sinks and massive activations already manifest. A systematic study of this setting is left for future work.

### F.2. Gating Mechanism

Trace back to previous literature, Bondarenko et al. (2023) found that introducing a gating mechanism into the attention score matrix can make the model more amenable to quantization. In a similar vein, Qiu et al. (2025) demonstrated that adding a gating mechanism to the attention output can mitigate both attention sink and massive activations. More intriguingly, they further showed that applying the gating mechanism only to the output of the value projection is sufficient to alleviate massive activations, while attention sink still persists. In the following, we following the implementation of gating mechanism in (Qiu et al., 2025), which is defined as

$$\boldsymbol{Y}' = \boldsymbol{g}\left(\boldsymbol{Y}, \boldsymbol{X}, \boldsymbol{W}_\theta, \sigma\right) = \boldsymbol{Y} \odot \sigma(\boldsymbol{W}_\theta \boldsymbol{X}),$$

where $\boldsymbol{Y}$ is the input to be modulated $\boldsymbol{X}$ is another input used to computing gating scores, which is usually the hidden states after pre-LN, $\boldsymbol{W}_\theta$ refers to the learnable parameters of gate, $\sigma$ is an activation function (i.e., sigmoid), and $\boldsymbol{Y}'$ is the gated output. We consider adding gating mechanism to value projection.

**Gating mechanism of value projection.** When a gating mechanism is introduced into the value projection of self-attention, the value computation becomes:

$$\boldsymbol{V} = \boldsymbol{W}_V \operatorname{LN}(\boldsymbol{h}_1^\ell) \Rightarrow \boldsymbol{g}(\boldsymbol{V}, \operatorname{LN}(\boldsymbol{h}_1^\ell), \boldsymbol{W}_\theta, \sigma) = \left(\boldsymbol{W}_V \operatorname{LN}(\boldsymbol{h}_1^\ell)\right) \odot \sigma\left(\boldsymbol{W}_\theta \operatorname{LN}(\boldsymbol{h}_1^\ell)\right).$$

To analyze how this gating operation affects the emergence of massive activations, we revisit the proof of Theorem 4.1, where we show that massive activations are required to control $\left\|\frac{\partial \boldsymbol{H}^{\ell+1/2}}{\partial \boldsymbol{h}_1^\ell}\right\|$. For notational simplicity, we restrict our discussion to the single-head attention case. By the chain rule, we have

$$\frac{\partial \boldsymbol{H}^{\ell+1/2}}{\partial \boldsymbol{h}_1^\ell} = \frac{\partial \boldsymbol{H}^{\ell+1/2}}{\partial \operatorname{LN}(\boldsymbol{h}_1^\ell)} \frac{\partial \operatorname{LN}(\boldsymbol{h}_1^\ell)}{\partial \boldsymbol{h}_1^\ell}.$$

Since the gate is applied only to the value projection, we focus on the gradient flowing through the value path. Without gating, this yields

$$\left\|\frac{\partial \boldsymbol{H}^{\ell+1/2}}{\partial \boldsymbol{h}_1^\ell}\right\| \lesssim \|\boldsymbol{W}_V\| \|\boldsymbol{A}_{1,:}\| \cdot \frac{1}{\|\boldsymbol{h}_1^\ell\|}.$$

When most attention mass concentrates on the first token, $\|\boldsymbol{A}1,:\|$ becomes large, which explains the necessity of massive activations for controlling the gradient norm.

After introducing the gating mechanism, the corresponding bound becomes

$$\left\|\frac{\partial \boldsymbol{H}^{\ell+1/2}}{\partial \boldsymbol{h}_1^\ell}\right\| \lesssim \left\|\frac{\partial \boldsymbol{g}(\boldsymbol{V}, \operatorname{LN}(\boldsymbol{h}_1^\ell), \boldsymbol{W}_\theta, \sigma)}{\partial \operatorname{LN}(\boldsymbol{h}_1^\ell)}\right\| \|\boldsymbol{A}_{1,:}\| \cdot \frac{1}{\|\boldsymbol{h}_1^\ell\|}.$$

To compute $\frac{\partial \boldsymbol{g}}{\partial \boldsymbol{x}}$, where $\boldsymbol{x} = \operatorname{LN}(\boldsymbol{h}_1^\ell)$, we have

$$\frac{\partial \boldsymbol{g}_i}{\partial \boldsymbol{x}} = \frac{(\boldsymbol{W}_V \boldsymbol{x})_i \cdot \sigma((\boldsymbol{W}_\theta \boldsymbol{x})_i)}{\partial \boldsymbol{x}} = (\boldsymbol{W}_V)_{i,:} \sigma((\boldsymbol{W}_\theta \boldsymbol{x})_i) + (\boldsymbol{W}_V \boldsymbol{x})_i \sigma'((\boldsymbol{W}_\theta \boldsymbol{x})_i)(\boldsymbol{W}_\theta)_{i,:},$$

where $\sigma'(z) = \sigma(z)(1 - \sigma(z))$. Let $\boldsymbol{s} = \sigma(\boldsymbol{W}_\theta \boldsymbol{x})$ and $\boldsymbol{s}' = \boldsymbol{s} \odot (1 - \boldsymbol{s})$. In matrix form, the Jacobian can be written as

$$\frac{\partial \boldsymbol{g}}{\partial \boldsymbol{x}} = \operatorname{diag}(\boldsymbol{s}) \boldsymbol{W}_V + \operatorname{diag}(\boldsymbol{s}') \operatorname{diag}(\boldsymbol{W}_V \boldsymbol{x}) \boldsymbol{W}_\theta.$$

Using this expression, we obtain the bound

$$\left\|\frac{\partial \boldsymbol{g}(\boldsymbol{V}, \operatorname{LN}(\boldsymbol{h}_1^\ell), \boldsymbol{W}_\theta, \sigma)}{\partial \operatorname{LN}(\boldsymbol{h}_1^\ell)}\right\| \lesssim \max_i |\boldsymbol{s}_i|.$$

Consequently,

$$\left\|\frac{\partial \boldsymbol{H}^{\ell+1/2}}{\partial \boldsymbol{h}_1^\ell}\right\| \lesssim \|\boldsymbol{A}_{1,:}\| \cdot \frac{\max_i |\boldsymbol{s}_i|}{\|\boldsymbol{h}_1^\ell\|}.$$

This bound implies that when $\max_i |\boldsymbol{s}_i|$ is sufficiently small, massive activations are no longer required to control the jacobian norm. Equivalently, the gating mechanism effectively reduces the constant $K_1$ in Theorem 4.1. As a result, we provide a theoretical explanation of the observation in (Qiu et al., 2025): adding a gating mechanism only to value projection can alleviate massive activations but attention sink still exists. In the following, we study another type of gating mechanism.

**Attention output gating.** When a gating mechanism is introduced into the output of each attention head, then the computation becomes

$$\sigma(\boldsymbol{W}_\theta \operatorname{LN}(\boldsymbol{h}_1^\ell)) \odot \boldsymbol{V} \operatorname{Softmax}(\boldsymbol{K}^\top \boldsymbol{Q}/\sqrt{d}).$$

As shown in (Bondarenko et al., 2023; An et al., 2025; Qiu et al., 2025), this gating or its variant allows flexible attention scores and can effectively eliminate attention sinks then eliminate massive activations.

### F.3. Layer normalization

In Section 4, we show that the position of massive activations largely depends on the LN structure. Massive activations in Pre-LN and post-LN have different position. In this section, we consider Transformer architecture without LN structure, motivated by recent works (Zhu et al., 2025). A Dynamic Tanh layer is defined as

$$\operatorname{DyT}(\boldsymbol{x}) = \boldsymbol{\gamma} \odot \tanh(\alpha \boldsymbol{x}) + \boldsymbol{\beta},$$

and the computation of self-attention becomes:

$$\boldsymbol{H}^{\ell+1/2} = \operatorname{Attn}\left(\operatorname{DyT}(\boldsymbol{H}^\ell)\right) + \boldsymbol{H}^\ell.$$

Then, following previous analysis framework, we have

$$\left\| \frac{\partial \boldsymbol{H}^{\ell+1/2}}{\partial \boldsymbol{h}_1^\ell} \right\| \lesssim \left\| \frac{\partial \boldsymbol{H}^{\ell+1/2}}{\partial \operatorname{DyT}(\boldsymbol{h}_1^\ell)} \right\| \left\| \frac{\partial \operatorname{DyT}(\boldsymbol{h}_1^\ell)}{\partial \boldsymbol{h}_1^\ell} \right\|.$$

To compute $\frac{\partial \operatorname{DyT}(\boldsymbol{x})}{\partial \boldsymbol{x}}$, where $\boldsymbol{x} = \boldsymbol{h}_1^\ell$, we have

$$\frac{\partial \operatorname{DyT}(\boldsymbol{x})_i}{\partial \boldsymbol{x}_j} = \frac{\partial \gamma_i \tanh(\alpha x_i) + \beta_i}{\partial x_i} = \begin{cases} \gamma_i \alpha \left(1 - \tanh^2(\alpha x_i)\right) & i = j, \\ 0 & i \neq j. \end{cases}$$

The Jacobian can be written as

$$\frac{\partial \operatorname{DyT}(\boldsymbol{x})}{\partial \boldsymbol{x}} = \operatorname{diag}\left(\alpha \boldsymbol{\gamma} \odot \left(1 - \tanh^2(\alpha \boldsymbol{x})\right)\right).$$

Consequently, we have

$$\left\| \frac{\partial \operatorname{DyT}(\boldsymbol{x})}{\partial \boldsymbol{x}} \right\| \lesssim e^{-2 \min_i |x_i|},$$

where we use the fact

$$\left| 1 - \tanh^2(x) \right| = \left| \operatorname{sech}^2(\boldsymbol{x}) \right| = \left( \frac{2}{e^x + e^{-x}} \right)^2 \lesssim e^{-2|x|}.$$

Then we have

$$\left\| \frac{\partial \boldsymbol{H}^{\ell+1/2}}{\partial \boldsymbol{h}_1^\ell} \right\| \lesssim \|\boldsymbol{W}_V\| \|\boldsymbol{A}_{1,:}\| \cdot e^{-2 \min_i |(\boldsymbol{h}_1^\ell)_i|}.$$

This bound implies that if $\min_i |(\boldsymbol{h}_1^\ell)_i| \sim \log(\frac{1}{\varepsilon})$, then the gradient can be well bounded and no massive activations are required to emerge even if attention sinks are present. See the empirical validation in Figure 6 of (Owen et al., 2025).

