# OpenReview forum: "Towards Understanding Massive Activations in Attention Sink Mechanism"
_ICML.cc/2026/Conference — ICML 2026 regular_

### Official Review · Reviewer_HbMg · 2026-02-20

**Soundness:** 3
**Presentation:** 3
**Significance:** 3
**Originality:** 3
**Overall Recommendation:** 4
**Confidence:** 2

**Summary:**

This work provide a theoretical analysis framework for two phenomenon observed in attention mechanism. (i) attention sink; (ii) massive activations. The authors argue that the common view “massive activations cause attention sink” is incomplete, and instead propose that the two phenomena co-exist to prevent excessive token mixing in self-attention. Empirically, the paper shows that attention sink can persist even when massive activations are suppressed, depending on where interventions are applied relative to normalization and how interventions propagate across layers.

**Compliance With Llm Reviewing Policy:**

Affirmed.

**Final Justification:**

Most of my concerns are resolved, only except the Theorem related concern.
The author rebuttal is very cleared and organized.

**Key Questions For Authors:**

Please refer to the weakness section above.

**Limitations:**

yes.

**Strengths And Weaknesses:**

Strength

- The empirical section is built around clear, targeted interventions (zeroing top‑k activations; scaling sink hidden states; zeroing sink value vectors) that directly test the causal relationship between massive activations and attention sink
- The paper is generally well structured: motivating experiments → theory → intervention analysis → implications for KV biases/gating/normalization. The narrative is easy to follow, and figures/tables are used to highlight key empirical takeaways
- Understanding the relationship between attention sinks and activation outliers is relevant for interpretability and for engineering interventions (quantization, KV-cache modification, gating), where outlier behavior is a known practical issue. The observation that suppressing massive activations can break sink structure via changes in value vectors, and that explicit value interventions can restore sink behavior, is a potentially useful mechanistic insight



Weakness

1. Theorem 5.1 relies on a strong stylized attention-sink assumption (essentially every non-sink token attends to the sink token with probability $1-\epsilon$, and discussion assumes sink behavior across heads), and the resulting bound contains large “absorbed constants” $K_1,K_2,K_3$ that are not estimated or validated empirically. This makes it harder to judge how predictive/tight the theory is beyond being a plausible explanatory argument
2. The sink metric and the theoretical discussion both depend on $T$ and $\epsilon$. How do the empirical conclusions change with different context lengths (short vs long) and different thresholds? (Could affect significance for long-context inference/streaming.)

2. Parts of the proof use additional conditions that are only briefly justified, e.g., Lemma B.1 states an extra assumption about high sparsity to reach a particular bound, which is plausible for outlier-dominated activations but not rigorously connected to typical LLM hidden-state statistics in the main text (Appendix B.4).
3. Empirical evaluation focuses mainly on sink metrics and perplexity under interventions. While this is appropriate for mechanism probing, the paper would be stronger with additional evidence that the proposed “over-mixing prevention” mechanism correlates with behavior on broader downstream tasks or long-context settings, not only perplexity on a few corpora

---

> ### Author Rebuttal · Authors · 2026-03-31
>
> Thank you for your objective review and valuable suggestions. Below, we respond to the comments in **Weaknesses (W)**, **Questions (Q)**.
>
> **W1: Theorem 5.1 relies on a strong ……**
>
> - **The $1-\epsilon$ Assumption:**  This assumption is well supported by empirical evidence showing that, in the attention sink regime, attention distributions are highly concentrated on sink tokens across layers and heads (as reflected in our $Sink_1^\epsilon$ measurements). Crucially, our framework does not require $\epsilon$ to be infinitesimally small; rather, it requires the term $\frac{\epsilon}{\sqrt{T-1}}$ to be small. This aligns seamlessly with practical scenarios: for large sequence lengths $T$, even a relatively large $\epsilon$ still results in the sink token dominating the attention weights. Furthermore, this assumption can be naturally relaxed. We can generalize it such that the $i$-th token attends to the sink with probability $1-\epsilon_i$. We can even introduce a threshold $K$ and an index set $I \subset \{1, 2, \dots, n\}$, allowing a subset of tokens ($i \in I$) to heavily attend to the sink ($1-\epsilon_i > K$), while others ($i \notin I$) do not ($1-\epsilon_i \leq K$).
>
> - **Cross-Head Behavior:** Regarding the assumption of sink behavior across all heads, this is not overly restrictive in practice. As shown in [1], the vast majority of attention heads naturally exhibit the attention sink pattern. Moreover, our analysis can be restricted to the subset of heads that display such behavior, while treating the Jacobian norms of the remaining heads as constants.
>
> - **The constants $K_1$, $K_2$, and $K_3$:** Regarding these unestimated constants, the primary objective of Theorem 5.1 is not to predict the exact numerical value of the Jacobian norm, but rather to formally establish its scaling behavior. Specifically, the bound demonstrates that the gradient norm decays inversely with $||h_1||_2$. For empirical validation of this trend, we respectfully refer the reviewer to our response to **W4** of Reviewer hC48. Furthermore, as discussed in Section 5, value projection gating can effectively reduce $K_1$. This claim is supported by the findings in [5], where gating scores are observed to be predominantly close to zero.
>
> **W2: The sink metric and the……**
>
> > We thank the reviewer for this insightful question. While $T=64$ and $\epsilon=0.3$ are common heuristics ([1], [2]), fixed $\epsilon$ becomes problematic for sequences with varying length. As shown in [1] and our Table 4, $Sink_1^\epsilon$ decreases with increasing $T$. This is because the attention weights decay toward $1/T$ as $T$ grows to infinity ([3,4]), making a fixed $\epsilon$ increasingly strict. Ideally, $\epsilon$ should scale with $T$. Since no standardized length-adjusted definition exists, we report results across a wide range of $\epsilon$ values (0.2–0.8) in Table 4, ensuring our conclusions are robust to the threshold choice.
>
>
> **W3: Parts of the proof ……**
>
> > We agree that the role of the high-sparsity assumption in Lemma B.1 should be made more explicit in the text. Importantly, we wish to clarify that this high-sparsity assumption is strictly required only for standard Layer Normalization, owing to its mean-centering operation. Because our work primarily investigates modern LLM architectures (e.g., Llama-2, Llama-3, and Mistral) that exclusively employ RMSNorm, our core theoretical bound in Theorem 5.1 is completely independent of this sparsity assumption. Since RMSNorm does not mean-center the activations, the bound holds rigorously without requiring any additional sparsity constraints. To address the reviewer's concern, we will update the Lemma B.1 in Appendix B.4 to explicitly delineate the RMSNorm proof (which requires no extra assumptions) from the layer normalization proof.
>
> **W4: Empirical evaluation focuses……**
>
> > We thank the reviewer for this highly constructive suggestion.  To address this, we have conducted an ablation study using Llama-2-7B across several standard benchmarks (500 samples). As shown in the table below, structurally removing the massive activations (w/o MA) leads to a severe degradation in downstream task performance compared to the standard model (w/ MA). We believe these results support our claims.
>
> |Benchmark| w/ MA|  w/o MA|
> |-----|-----|-----|
> |HellaSwag|70 | 28|
> |ARC-e|79  | 28|
> |ARC-c|43  | 24|
> |PIQA|79  | 49|
> |winogrande|68 | 48|.
>
> Furthermore, we will include additional evaluations on long-context tasks using the RULER benchmark in the revised manuscript. We greatly appreciate the reviewer's guidance in expanding the scope of our empirical validation.
>
> **Reference**
>
> [1] WHEN ATTENTION SINK EMERGES  IN LANGUAGE MODELS: AN EMPIRICAL VIEW.
>
> [2] Why do LLMs attend to the first token?
>
> [3] Softmax is not Enough (for Sharp Size Generalisation).
>
> [4] Limitations of Normalization in Attention Mechanism.
>
> [5] Gated Attention for Large Language Models: Non-linearity, Sparsity, and Attention-Sink-Free.

---

> > ### Author Rebuttal · Reviewer_HbMg · 2026-04-05
> >
> > Thank you for the detailed rebuttal. The additional clarifications on the RMSNorm-vs-LayerNorm assumption, the robustness across sink thresholds, and the added downstream ablations are helpful, and they address a substantial part of my concerns.
> >
> > My remaining question is mainly about the practical predictive value of Theorem 5.1. I understand that the theorem is intended to capture the scaling behavior rather than predict exact Jacobian magnitudes. However, since the explanation relies on several absorbed constants and a stylized sink assumption, it would still be helpful if the final version could more clearly discuss how one should interpret the theorem empirically: for example, under what regimes it should be expected to be qualitatively predictive, and where it is mainly best understood as a mechanistic explanatory argument rather than a quantitatively tight bound.
> >
> > This would make the connection between the formal result and the intervention evidence clearer.
> >
> > I have raised my score.

---

> > > ### Author Response · Authors · 2026-04-05
> > >
> > > We sincerely thank you for the positive feedback, the increased score, and this highly insightful suggestion.
> > >
> > > We will clarify, both in the main text and in Appendix B.2, that Theorem 5.1 is not intended to provide a quantitatively tight bound on the exact Jacobian magnitudes in a live model. Because the constants ($K_1, K_2, K_3$) absorb complex weight dynamics and the proof relies on worst-case perturbation bounds, the resulting numerical magnitude is naturally loose. Furthermore, while the stylized sink assumption effectively captures the essence of the phenomenon, it inherently contributes to this lack of tightness. As such, the theorem is best understood as a mechanistic explanation for the inverse scaling relationship between activation norms and gradient sensitivity. It formally proves why the combination of Softmax-induced attention concentration and RMSNorm necessitates massive activations to prevent over-mixing, rather than predicting the exact numerical threshold at which they emerge. Finally, we will add a discussion outlining potential methodological refinements (such as incorporating more realistic and nuanced assumptions, like those in our rebuttal) that could be pursued to make this bound quantitatively predictive.
> > >
> > > Thank you again for helping us improve the clarity and impact of this work!

---

### Official Review · Reviewer_VGQg · 2026-03-12

**Soundness:** 3
**Presentation:** 2
**Significance:** 3
**Originality:** 4
**Overall Recommendation:** 4
**Confidence:** 3

**Summary:**

This paper investigates the relationship between two recently observed phenomena in Transformer-based large language models: massive activations and the attention sink mechanism. The authors challenge the prevailing interpretation that massive activations are the primary cause of attention sink, and instead propose a more nuanced view where both mechanisms jointly regulate token mixing in self-attention. Specifically, attention sink suppresses mixing among non-sink tokens, while massive activations suppress mixing between sink and non-sink tokens. The paper further provides theoretical analysis explaining how normalization placement affects massive activations and empirically studies interventions such as removing sink token value vectors. The results aim to provide a mechanistic understanding of how these phenomena interact in Transformer architectures.

**Compliance With Llm Reviewing Policy:**

Affirmed.

**Key Questions For Authors:**

Q1. How consistent are the observed interactions between massive activations and attention sink across different model architectures, model sizes, and training setups? In particular, have the authors verified whether the same behavior appears in larger or more recent LLMs? Demonstrating such consistency would significantly strengthen the generality of the proposed explanation.

Q2. Given that the paper studies the interaction between attention sink and value vectors, could manipulating or redistributing KV weights help correct “sink heads” that disproportionately capture attention and potentially contribute to hallucination? If so, could the proposed insights be leveraged to reduce hallucination or improve factual reliability in LLMs?

**Limitations:**

yes

**Strengths And Weaknesses:**

Strengths

1. The attention sink mechanism has recently become an actively discussed phenomenon in the community, yet its underlying causes remain unclear and multiple explanations coexist. This paper presents an interesting exploratory interpretability study that investigates the relationship between attention sink and massive activations, offering a novel perspective on how these phenomena interact in Transformer models. Such mechanistic analysis is timely and contributes to a deeper understanding of LLM internal dynamics.

2. The paper includes theoretical analysis regarding the role of normalization placement as well as empirical intervention studies, which together aim to support the proposed interpretation.

Weaknesses

1. While the paper proposes an interesting conceptual explanation for the interaction between massive activations and attention sink, the empirical evidence appears relatively limited. The experiments mainly focus on a small set of intervention analyses, and it remains unclear whether the observed behaviors generalize across different model scales, architectures, or training regimes. Additional experiments across more models and settings would strengthen the validity and generality of the proposed mechanism.

2. The paper would benefit from additional polishing in terms of writing and formatting. For example, the usage of punctuation in section and subsection titles is inconsistent (some titles end with periods while others do not). There are also minor writing issues such as inconsistent abbreviation usage—for instance, “Large Language Models” is first introduced without abbreviation but abbreviated as “LLMs” only later in the same context. Addressing these presentation inconsistencies would improve the overall clarity and professionalism of the paper.

---

> ### Author Rebuttal · Authors · 2026-03-31
>
> Thank you for your supportive review and valuable suggestions. Below, we respond to the comments in **Weaknesses (W)**, **Questions (Q)**.
>
> **W1 (Q1): While the paper proposes …….**
>
> > We thank the reviewer for highlighting the importance of generality.
> - **Model Scales:** In this work, we primarily focused on the Llama-2, Llama-3, Llama-3.1, and Mistral families, as they represent the most widely adopted open-weight models in current literature [1, 2]. However, we agree that validating our findings across a wider spectrum of model sizes is a valuable addition. For the revised manuscript, we will extend our empirical analysis to include the Gemma, OPT, Pythia, and Qwen families, spanning parameter counts from 14M to 13B. We appreciate this constructive suggestion.
>
> - **Model Architectures:** The pre-LN structure coupled with Softmax self-attention is the dominant architecture today, which motivated our primary empirical focus. Crucially, however, our theoretical framework is not limited to this specific setup. As detailed in Section 5 and Appendix E , our theory provides a unified mechanistic explanation for behaviors observed in models utilizing post-LN , Dynamic Tanh (DyT) , KV biases , gating mechanisms , and unnormalized activation functions. Since these different components have been widely studied in existing literature, our work establishes the first mathematical foundation that explains the diverse empirical observations.
>
> - **Training Regimes:** We agree that training regimes play a significant role in the initial formation of both attention sinks and massive activations. Because our theoretical framework is strictly grounded in the mechanics of token mixing at inference, analyzing the complex developmental dynamics of the training phase falls outside the immediate scope of this paper. Nonetheless, our mechanistic insights are highly complementary to recent empirical investigations into training. For instance, [1] studies how hyperparameters (such as learning rate, weight decay, and batch size) affect attention sink formation, while a concurrent work [3] explores the emergence of massive activations during training.
>
> **W2: The paper would benefit from …….**
>
> > We sincerely thank the reviewer for their careful reading and for pointing out these presentation issues, and we apologize for the oversights. As also noted by reviewer hC48, our paper presentation needs improvement. In the revised manuscript, we will undertake a thorough proofreading pass to ensure clarity and professionalism. Specifically, we will unify the punctuation across all section, subsection, and paragraph headings to ensure a consistent style throughout the paper. Besides, we will ensure that all terms, such as "Large Language Models (LLMs)", are properly defined at their first occurrence and used consistently thereafter. We greatly appreciate the reviewer's attention to detail, which will undoubtedly improve the final quality of the paper.
>
> **Q2: Given that the paper……**
>
> > We thank the reviewer for this highly insightful and thought-provoking question. To the best of our knowledge, several recent works have drawn connections between the attention sink phenomenon and model hallucination ([4], [5], [6]). These studies suggest that the emergence of hallucinations is strongly correlated with attention sinks, and that calibrating these attention weights can mitigate such factual errors. In our work, we demonstrate that improper modifications to the value vectors of the sink token disrupt the attention sink structure , whereas explicitly setting these value vectors to zero can actually recover it. However, the precise functional roles of the sink token's keys and values remain an open question. Are they solely dedicated to attracting the majority of attention weights, or do they carry other semantic purposes? We agree that fully unraveling their distinct functional roles is a critical next step. In the revised manuscript, we will expand our discussion to explicitly bridge our value-vector intervention analysis with potential strategies for hallucination mitigation.
>
>
>
> **Reference**
>
> [1] When Attention Sink Emerges in Language Models: An Empirical View.
>
> [2] Massive Activations in Large Language Models.
>
> [3] The Spike, the Sparse and the Sink: Anatomy of Massive Activations and Attention Sinks.
>
> [4] OPERA: Alleviating Hallucination in Multi-Modal Large Language Models via Over-Trust Penalty and Retrospection-Allocation.
>
> [5] Seeing Clearly by Layer Two: Enhancing Attention Heads to Alleviate Hallucination in LVLMs.
>
> [6] SAGE: Sink-Aware Grounded Decoding for Multimodal Hallucination Mitigation.

---

### Official Review · Reviewer_am8j · 2026-03-13

**Soundness:** 3
**Presentation:** 3
**Significance:** 3
**Originality:** 3
**Overall Recommendation:** 5
**Confidence:** 2

**Summary:**

The paper provides a mechanistic exploration of the relationship between **massive activations** (hidden states with outlier magnitudes) and the **attention sink** phenomenon in Large Language Models (LLMs). The authors challenge the prevailing view that massive activations are the primary driver of attention sinks, demonstrating that the two phenomena are decoupled in their formation. Through theoretical analysis based on the concept of **over-mixing**, they argue that massive activations act to stabilize representations and suppress excessive mixing between sink and non-sink tokens. The study also explains how architectural choices like Pre-LN, KV-biases, and gating influence these phenomena, supported by intervention analyses on models such as Llama-2 and Llama-3.

**Compliance With Llm Reviewing Policy:**

Affirmed.

**Final Justification:**

i find this paper of solid methodology and high impact for the field, why is why i grade it 5.

**Key Questions For Authors:**

Given that the current study focuses heavily on the [BOS] token as the primary sink, have you conducted analyses on secondary sinks or delimiter tokens to ensure the generalizability of your theoretical framework?

Could you provide a more detailed and descriptive caption for Figure 3 to improve its interpretability?

Would you consider moving the discussion regarding the ineffectiveness of deeper layers from the appendix to the main manuscript, given its strong conceptual link to your theory of over-mixing?

How do your findings regarding massive activations as a stabilization mechanism inform the design of future Transformer architectures? Beyond the formal proofs provided, could you offer more mechanistic intuition and prescriptive advice on how your findings regarding massive activations can be used to design more optimal future architectures that are inherently free from outliers?

**Limitations:**

The research provides significant formal insights into the relationship between massive activations and attention sinks.

However, the contribution would likely attract greater attention from the community if the formal proofs were accompanied by more mechanistic intuition and prescriptive advice.

Specifically, the paper would benefit from a discussion on how these findings can be directly applied to design more optimal future architectures.

**Strengths And Weaknesses:**

### **Soundness**

*
**Strength:** The authors provide a rigorous theoretical framework.


*
**Strength:** The intervention experiments (setting activations to zero before vs. after Layer Norm) effectively isolate the role of normalization in the attention sink mechanism.


*
**Weakness:** The study focuses heavily on the **beginning-of-sequence [BOS]** token as the primary sink; a more detailed analysis of secondary sinks or delimiter tokens mentioned in the related work would strengthen the generalizability of the findings.



### **Presentation**

*
**Strength:** The paper is well-organized, successfully bridging empirical observations with mathematical proofs.


*
**Weakness:** Caption in Figure 3 is too sparse.


*
**Weakness:** The links to ineffectiveness of deeper layers are relegated to the appendix, when they should probably be in main paper.



### **Significance**

*
**Strength:** Disentangling massive activations from attention sinks is a significant contribution to the mechanistic interpretability of Transformers.


### **Originality**

*
**Strength:** The conceptualization of massive activations as a stabilization mechanism seems to be a novel perspective.

---

> ### Author Rebuttal · Authors · 2026-03-30
>
> We sincerely thank the reviewer for their time, constructive, positive feedback, and insightful comments. Below, we respond to the comments in **Weaknesses (W)**, **Questions (Q)** and **Limitations (L)**
>
> **W1 (Q1): The study focuses heavily on the beginning-of-sequence [BOS] token as the primary sink; a more detailed analysis of secondary sinks or delimiter tokens mentioned in the related work would strengthen the generalizability of the findings.**
>
> > We thank the reviewer for this constructive feedback. As noted in Section 5, our theoretical framework naturally extends to scenarios involving multiple sink tokens, which aligns with various empirical observations in the literature. We primarily focus on the initial token because the formation mechanisms of secondary sinks remain an open problem and are significantly less understood. While several works (e.g., [1], [2]) have empirically observed secondary sinks, a theoretical explanation for their formation is still lacking. Our framework firmly establishes that if secondary sinks appear, massive activations will correspondingly emerge. However, predicting exactly why and when these secondary sinks form is beyond the scope of our paper. As a result, we have explicitly framed the formation of secondary sinks as a promising direction for future research. In the revised manuscript, we will add a discussion about secondary sinks in the appendix. Many thanks again.
>
> **W2 (Q2): Caption in Figure 3 is too sparse. / Could you provide a more detailed and descriptive caption for Figure 3 to improve its interpretability?**
>
> > We agree with the reviewer that the original caption for Figure 3 was too brief, and we thank the reviewer for pointing this out. As detailed in our analysis before Figure 3, we theoretically predicted that in LLMs with a post-LayerNorm (post-LN) structure, massive activations should emerge specifically at the input to the second layer normalization module ($H^\*$). Figure 3 empirically validates this prediction by plotting $H^\*$ of the [SEP] token across layers in BERT using the prompt: “This is a simple test sentence for BERT.” As expected, massive activations clearly appear in $H^*$. For comparison, we also plot the standard hidden states $H^\ell$ across layers, where no such massive activations are observed. In the revised manuscript, we will expand this caption to include these details, ensuring the figure is fully self-contained.
>
> **W3 (Q3): The links to ineffectiveness of deeper layers ……**
>
> > We thank the reviewer for their careful reading and valuable suggestion. Currently, due to strict page limits, Section 5 briefly directs readers to Appendix C for a detailed discussion on the ineffectiveness of deeper layers. However, we agree that this concept is central to our narrative and deserves more prominence. In light of your feedback, and aligning with Reviewer hC48's suggestions to improve the paper's overall presentation, we are comprehensively reorganizing the main text. By streamlining other sections, we will free up the necessary space to integrate the core discussion regarding the ineffectiveness of deeper layers directly into the main manuscript."
>
> **Q4 (L): How do your findings ……**
>
> > We thank the reviewer for this forward-looking question. Our framework explains why massive activations emerge, which is known to pose challenges for quantization [1]. Building on our findings, we propose three architectural directions to prevent them:
>
> - **Preventing attention sinks**: Theorem 5.1 demonstrates that massive activations arise to suppress the over-mixing caused by attention sinks. Utilizing alternative attention functions (that do not enforce a strict sum-to-one constraint) can eliminate attention sinks entirely, preempting the need for massive activations ([1]). Modifying residual connection or LN modules to handle token routing may achieve a similar effect ([2]).
>
> - **Strategic gating mechanisms:** As analyzed in Section 5, gating mitigates massive activations in two distinct ways : (i) attention probability gating eliminates the sink entirely, while (ii) value projection gating directly suppresses sink/non-sink mixing along the value path, which mitigates massive activations while preserving the sink. This theoretically guides the placement of gates ([3]).
>
> - **Alternative layer normalization:** Theorem 5.1 shows that standard LN necessitates massive activations due to its $||h_1^l||_2$ decay factor. Adopting normalization modules with higher-order decay—such as Dynamic Tanh (DyT), which induces an exponential decay of $exp(-||h_1^l||_2)$—naturally circumvents the need for extreme activation norms.
>
>
> **Reference**
>
> [1] Quantizable Transformers: Removing Outliers by Helping Attention Heads Do Nothing.
>
> [2] Attention Residuals.
>
> [3] Gated Attention for Large Language Models: Non-linearity, Sparsity, and Attention-Sink-Free.

---

> > ### Author Rebuttal · Reviewer_am8j · 2026-04-03
> >
> > Thank you for this strong paper and strong rebuttal, my concerns have been addressed.

---

> > > ### Author Response · Authors · 2026-04-04
> > >
> > > We greatly appreciate the time and effort you invested in reviewing our paper and considering our rebuttal. Your thoughtful and insightful feedback has significantly improved both the quality and clarity of our work. Thank you for your support of our paper!

---

### Official Review · Reviewer_hC48 · 2026-03-18

**Soundness:** 2
**Presentation:** 1
**Significance:** 3
**Originality:** 3
**Overall Recommendation:** 4
**Confidence:** 4

**Summary:**

This paper studies the relationship between attention sink and massive activations in large language models.
The main claim is that massive activations are not the primary cause of attention sink, but instead attention sink and massive activations work together to reduce excessive token mixing in self-attention.
This paper theoretically and empirically shows that attention sink suppresses mixing among non-sink tokens, while massive activations suppress mixing between the sink token and the remaining tokens.

**Compliance With Llm Reviewing Policy:**

Affirmed.

**Final Justification:**

My concerns have been resolved, so I raise my score.

**Key Questions For Authors:**

1. Could you directly measure layerwise quantities such as |dH^(l+1/2) / dh_1^l|| and ||dH^(l+1/2) / dh_j^l|| for j > 1
in real models, and show that they scale with 1 / ||h_1^l|| and eps as the theorem suggests?
 Without such measurements, it is hard to know whether the theorem captures the mechanism in practice.
2. Colud you disentangle the effects of h = 0 on q, k, and v? For example, after applying h = 0, could you patch back only k_1, only v_1, or only LN(h_1) and compare sink rate and performance? That would help distinguish value contamination from q-k geometry.
3. Could you track cos(q_t, k_1), cos(q_t, k_j) , and pairwise cosine similarities among non-sink tokens before and after interventions?

**Limitations:**

See Weaknesses and Questions.

**Strengths And Weaknesses:**

Strengths

1. The paper shows that the two phenomena play complementary roles in controlling token mixing.
2. Theorem 5.1 gives a decomposition of the Jacobian upper bound into three interpretable pieces. The paper then uses this decomposition to explain several empirical observations, including why strong sink suppresses non-sink mixing, why large sink-token norms can stabilize sink-mediated interactions, and why design changes such as KV biases and gating can reduce massive activations while preserving sink behavior.
3. The paper perturbs the first token and shows that, without massive activations, the last-token representation becomes much more sensitive, while the effect of massive activations is substantially reduced, which the theory show.

Weaknesses
1. The paper does not use a direct mixing metric. The Jacobian does encode sensitivity, and off-diagonal Jacobian blocks correspond naturally to cross-token influence.
However, the full spectral norm of the whole Jacobian also contains same-token sensitivity and residual-path effects. This is not so fatal, but it should be stated more carefully.
2. The fact that v = 0 tends to preserve current-layer sink is not surprising on its own.
In standard attention, attention weights are determined by Q and K, not V, so setting the value vector to zero should not directly destroy the current-layer attention pattern.
This result is consistent with the authors’theory, but it is not uniquely diagnostic of it.
3. The alpha h intervention is also partly explained by the RMSNorm algebra.
Since pre-LN RMSNorm is approximately scale-invariant, scaling the hidden state can leave the normalized representation nearly unchanged.
Thus, the fact that alpha h does not always destroy sink is at least partly a direct consequence of the normalization formula, not necessarily evidence for the thoery proposed by the paper.
4.The degradation under h = 0 does not uniquely support the paper’s explanation.
This intervention can be just generically destructive, i.e., zeroing the top-5 massive activations of the sink token perturbs the source hidden state itself, which simultaneously affects q, k, and v.
5. The paper explicitly says, "We hypothesize that an improper value vector for the sink token disrupts this negative cosine-similarity structure among non-sink tokens."
The changes in sink rate, value norms, and perplexity do not directly measure whether the relevant cosine-geometry among non-sink tokens is actually destroyed under the claimed interventions.
6.The theory is about the Jacobian norm and its decomposition;however, the experiments mostly use sink rate, perplexity, and representation-distance proxies.
The perturbation experiment is helpful, but the paper does not directly measures ||dH^(l+1/2) / dh_1^l||, ||dH^(l+1/2) / dh_j^l|| for j > 1, or anything close to a decomposition into the theorem’s first and second terms.

---

> ### Author Rebuttal · Authors · 2026-03-30
>
> We sincerely thank the reviewer for the time, constructive feedback, and insightful comments. Below, we respond to the comments in **Weaknesses (W)** and **Questions (Q)**
>
> **W1: The paper does not ……**
> > We sincerely thank the reviewer for this insightful comment. We completely agree that this nuance should be stated more carefully. While the spectral norm of the Jacobian serves as a macroscopic upper bound that inherently encompasses same-token transformations, cross-token sensitivity, and residual-path effects, controlling this overall norm effectively bounds the cross-token mixing between sink and non-sink tokens. To clarify this in the revision, we will add a dedicated discussion distinguishing the specific contributions from: (i) cross-token mixing $||\frac{\partial H_{:,i}^{l+1/2}}{\partial H_{:,1}^l }||$, $i\neq 1$, (ii) same-token transformation $||\frac{\partial H_{:,1}^{l+1/2}}{\partial H_{:,1}}||$, and (iii) residual path. this decomposition follows naturally from our existing theoretical framework and can be explicitly demonstrated using our computation of $J_{ij} $(detailed on page 18).
>
> **W2: The fact that v = 0……**
> > We thank the reviewer for this sharp observation. However, the metric $Sink_1^\epsilon$ aggregates attention sink behavior across all layers and heads. Therefore, Tables 1 and 4 demonstrate that setting $v=0$ restores the attention sink across the entire depth of the network, not just the current layer. This experiment was not designed as direct proof of our theorem, but rather to systematically challenge the prevailing hypothesis that massive activations drive the attention sink. Our logic is as follows: 1. Zeroing out massive activations ($h=0$) decreases $Sink_1^\epsilon$, which seemingly supports the existing literature. 2. However, modifying the hidden state inevitably corrupts subsequent $k_1$ and $v_1$. 3. By explicitly setting $v_1=0$, $Sink_1^\epsilon$ recovers. This confirms that the sink's disappearance under the $h=0$ condition is mainly caused by the corrupted $v_1$, rather than the absence of massive activations themselves. We further disentangle the specific effects of $h=0$ on $k_1$ and $v_1$ in our detailed response to **W3 (Q2)** below.
>
> **W3(Q2): The alpha h intervention is …….**
> >  We introduced the $\alpha h$ experiment in Section 4 ("Motivating Examples") not as proof for our theory, but as an exploratory observation. Because RMSNorm is approximately scale-invariant, massive activations are not strictly necessary to produce the same normalized output. This specific finding is exactly what motivated our theoretical analysis to uncover their true functional role. As discussed in our response to **W2**, we hypothesized that the corrupted $v_1$ is the primary culprit behind the degradation of $Sink_1^\epsilon$. Following your valuable suggestion, we disentangled the effects of $k_1$ and $v_1$ after applying $h=0$. Our new ablation confirms that patching back the corrupted $v_1$ yields the largest decrease of both $Sink_1^\epsilon$ and perplexity.
>
> ||$Sink_1^\epsilon$| Perplexity |
> |-----|-----|-----|
> |original | 93.16 | 5.34|
> |$h=0$ | 6.05 | inf|
> |Patch back $k_1$| 67.76  |139 |
> | Patch back $v_1$|  26.27| inf |
>
>
> **W4 (Q1, Q3): The paper explicitly says ……**
> > We sincerely thank the reviewer for this constructive suggestion. We agree that directly measuring internal geometry and Jacobian norms strengthens the link between theory and experiments. Following your suggestion, we conducted these measurements on Llama2-7B, and the results strongly support our claims.
>
> - **Tracking Cosine-Similarity Geometry**
> ||original| $h=0$| $h=v=0$|
> |-----|-----|-----|-----|
> |$cos(k_1, q_{t\neq 1})$|0.5425 | -0.0474|0.2173 |
> |$cos(k_{t\neq 1}, q_{t\neq 1})$|-0.2319  | 0.0417| -0.1593|.
>
> Setting $h=0$ catastrophically disrupts the attention geometry by flipping the similarity signs. Removing the corrupted value vector ($h=v=0$) successfully restores the critical negative similarity among non-sink tokens, confirming our hypothesis regarding the cosine geometry.
>
> - **Jacobian Norms**
>
> We use $\alpha$ to denote the scaling factor assigned with $h_1$ and $\epsilon$ denote the attention weights of the first token. (They are independent although presented in the same table). We use $J_1$ to denote $||\frac{\partial H^{l+1/2}}{\partial h_1}||$ and $J_2$ to denote $||\frac{\partial H^{l+1/2}}{\partial h_i}||$.
>
> |$\alpha $|$J_1$ | $\epsilon$|$J_2$|
> |-----|-----|-----|-----|
> |0.01|18.4|  0.9| 5.02|
> |0.02|13.5| 0.8| 6.55|
> |0.05|11.1| 0.7 | 7.9|
> |0.1| 8.9| 0.6 |9.6 |
> |0.2|6.9| 0.5 | 10.8|
> |0.5|4.8| 0.4| 12.2|
> |1|4.0| 0.3| 14.0|
> |2|2.8| 0.2| 16.1|
> |5|1.8| 0.1| 17.0|.
>
> These empirical measurements clearly demonstrate the mathematical bounds established in our theory: $J_1$ strictly shrinks as the norm of the sink token increases, while $J_2$ diminishes alongside a increase in the first token attention weight $\epsilon$.

---

> > ### Author Rebuttal · Reviewer_hC48 · 2026-04-03
> >
> > Thank you for your response. My concerns have been resolved, so I will raise my score.

---

> > > ### Author Response · Authors · 2026-04-03
> > >
> > > We sincerely appreciate your encouraging response and for initiating such a thought-provoking discussion.
> > >
> > > Your suggestions—carefully disentangling token mixing, measuring the Jacobian norm, and tracking the cosine geometry of the QK vectors—are extremely helpful. They not only provide valuable directions for further validating our theory and hypotheses, but also help clarify our overall understanding. We plan to extend these analyses to a broader range of model architectures and scales, and incorporate the results into the final version.
> > >
> > > Many thanks again for your time and insightful feedback throughout this process.

---

### Decision · Program_Chairs · 2026-04-30

**Decision:**

Accept (regular)

**Comment:**

The paper revisits the relationship between massive activations and attention sinks in LLMs. The authors argue that the two phenomena play complementary roles in controlling token mixing: attention sinks suppress mixing among non-sink tokens, while massive activations suppress mixing between sink and non-sink tokens. The analysis is supported by several experiments.

Reviewers found the perspective interesting. Main concerns were:

* the experiments measure sink rate and perplexity rather than the cross-token Jacobian quantities that the theory actually bounds
* the results may not generalize beyond a few model families,
* Thm 5.1 relies on strong assumptions.

The authors addressed some of these in the rebuttal. hC48 and HbMg raised their scores after the rebuttal.

I think the theoretical contribution could be improved. The proof of Thm 5.1 is technically standard:

* it decomposes the Jacobian via the chain rule, applies sub-multiplicativity of the spectral norm, and relies on standard bounds for the softmax and RMSNorm Jacobians (Lems B.1 and B.2), which follow patterns already established in literature (Noci et al., 2022).

* The $1/||h_1^\ell||_2$ dependence comes directly from the RMSNorm Jacobian bound rather than from a new argument specific to the massive-activation phenomenon.

I also agree with Reviewer HbMg that the assumptions are quite strong. The theorem assumes a stylized attention-sink pattern where every non-sink token attends to the sink with probability $1-\varepsilon$ across all heads, and the constants $K_1, K_2, K_3$ absorb model parameters and other token norms without being estimated. As a result, the bound is best read as an explanatory scaling argument rather than a tight or predictive bound.